# A definition of primary operators in $J\overline{T}$-deformed CFTs

**Monica Guica**

Université Paris-Saclay, CNRS, CEA,
Institut de Physique Théorique, 91191 Gif-sur-Yvette, France

## Abstract

$J\overline{T}$ - deformed CFTs provide an interesting example of non-local, yet UV-complete two-dimensional QFTs that are entirely solvable. They have been recently shown to possess an infinite set of symmetries, which are a continuous deformation of the Virasoro-Kac-Moody symmetries of the seed CFT. In this article, we put forth a definition of primary operators in $J\overline{T}$ - deformed CFTs on a cylinder, which are singled out by having CFT-like momentum-space commutation relations with the symmetry generators in the decompatification limit. We show - based on results we first derive for the case of $J^1 \wedge J^2$ - deformed CFTs - that all correlation functions of such operators in the $J\overline{T}$ - deformed CFT can be computed exactly in terms of the correlation functions of the undeformed CFT and are crossing symmetric in the plane limit. In particular, two and three-point functions are simply given by the corresponding momentum-space correlator in the undeformed CFT, with all dimensions replaced by particular momentum-dependent conformal dimensions. Interestingly, scattering amplitudes off the near-horizon of extremal black holes are known to take a strikingly similar form.



# 1 Introduction

A long-standing challenge in holography has been to find the microscopic description of *generic* extremal and non-extremal black holes, whose near-horizon region does not contain an AdS$_3$ factor. One can obtain important clues into the nature of the holographic dual by studying the asymptotic symmetries of the near-horizon backgrounds of interest [1], as well as scattering amplitudes [2,3], which in principle give access to the symmetries [4] and, respectively, the correlation functions of the dual theory.

To date, the most progress has been made in the case of extremal black holes, whose near-horizon region universally contains [5] a factor known as warped AdS$_3$: a deformation of AdS$_3$ that preserves $SL(2,\mathbb{R}) \times U(1)$ isometry. The asymptotic symmetries of this space-time enhance the $U(1)$ factor to one copy of the Virasoro algebra, leading to the so-called Kerr/CFT proposal [6], which states that the near-horizon dynamics of extremal black holes is described by a chiral half of a two-dimensional CFT, where the chirality is due to taking the strict extremal limit [7].

The study of scattering off the near-extremal black hole geometry naively appears to corroborate this claim, since it leads to a (scalar) momentum-space two-point function of the form [8]

$$\mathcal{G}_{T_{L,R}}(p,\bar{p}) \sim T_L^{2h(\bar{p})-1} e^{-\frac{p}{2T_L}} \left| \Gamma\left( h(\bar{p}) + \frac{ip}{2\pi T_L} \right) \right|^2 \times T_R^{2h(\bar{p})-1} e^{-\frac{\bar{p}}{2T_R}} \left| \Gamma\left( h(\bar{p}) + \frac{i\bar{p}}{2\pi T_R} \right) \right|^2, \quad (1.1)$$

which precisely corresponds to the momentum-space two-point function in a two-dimensional CFT at left and, respectively, right-moving temperatures $T_{L,R}$. Similar results hold for higher-point functions [9].

This conclusion may nevertheless be a bit too fast, because the operator dimensions that appear in the above formula, which are extracted from the solution to the scalar wave equation in the near-horizon geometry, depend explicitly on the momentum, $\bar{p}$, along the $U(1)$ direction. This fact immediately implies that the "CFT" in the Kerr/CFT correspondence cannot be a standard, local CFT[1]. Indeed, a detailed study of the near-horizon geometry [13] reveals that the holographic dual is instead a two-dimensional analogue of a dipole theory [14], obtained by deforming the CFT by a finely-tuned set of irrelevant operators that preserve the left $SL(2,\mathbb{R})$ symmetry and lead to a UV-complete theory. This structure of the irrelevant deformation implies that the resulting theory - sometimes called a "dipole CFT" - is local and conformal on the left, but non-local on the right. Operators in these theories are best described in a mixed position - momentum basis, and the left-moving piece of their correlation functions has a form

---

[1]Consequently, the asymptotic symmetry results of [6] (or, more rigorously, [10]) should not be interpreted as suggesting that the dual theory is a standard CFT, but rather that non-local theories can posses Virasoro symmetry, a fact that was recently proven in [11,12].

dictated by the $SL(2,\mathbb{R})_L$ conformal symmetry; however, the left conformal dimensions generically depend on the right-moving momentum [15].

The fact that the dual theory is non-local raises some immediate questions regarding (1.1). While the left-moving piece of this correlator is meaningful - since it corresponds to the Fourier transform of a local, conformal two-point function (albeit with a dimension that depends on the momentum along the other direction) - its right-moving piece, which resides entirely on the non-local direction, appears to have no intrinsic meaning, since it can always be rescaled by an arbitrary function of $\bar{p}$ [16]. The question that we would like to address in this article is: to what extent is an expression of the form (1.1) in such a non-local theory meaningful?

That this correlator may actually be meaningful is suggested by the example of dipole-deformed field-theories, whose non-locality is extremely constrained by a star product structure. Operators in this theory and the allowed counterterms must respect this structure, which effectively makes the theory renormalizable [17], if the original QFT was so. Thus, in presence of additional structures that constrain the non-locality, one may hope to be able to assign an unambiguous meaning to an expression such as (1.1).

The main goal of this article is to identify such additional structures for the case of two-dimensional non-local QFTs that may model the microscopic physics of generic extremal black holes. Specifically, we will be working with the example of $J\bar{T}$ - deformed CFTs [18] - a class of *universal* irrelevant deformations of two-dimensional CFTs by an operator that is bilinear in the stress tensor and a $U(1)$ current, which lead to a UV-complete QFT. These theories belong to the more general class of Smirnov-Zamolodchikov irrelevant current-current deformations [19], of which the $T\bar{T}$ deformation [19,20] is a particularly rich and interesting example [21–24]. While $J\bar{T}$ - deformed CFTs are not exactly a model for the Kerr/CFT correspondence because the deformation is double-trace - and thus corresponds to AdS$_3$ with mixed boundary conditions [25] - a single-trace variant [26,27] of this deformation is. Notwithstanding, $J\bar{T}$ - deformed CFTs do appear to posses the correct non-local QFT structures that are relevant for the Kerr/CFT correspondence, in addition to being highly tractable. The concrete question that we would like to address in this article is whether a formula such as (1.1) can be made sense of in the context of $J\bar{T}$ - deformed CFTs.

This question has two aspects. The first is that one should find a basis of operators for which the correlation functions are expected to take a simple form. In usual CFTs, these are local operators - in fact, primary operators, if we want the higher-point functions to also be nicely constrained. In $J\bar{T}$ - deformed CFTs, the primary constraint is easily imposed on the local left-moving side and yields correlation functions that are consistent with $SL(2,\mathbb{R})$ invariance, the only change being that the left conformal dimensions and $U(1)$ charges are shifted from their undeformed values $\tilde{h}, \tilde{q}$ by a momentum-dependent contribution [28]

$$h(\bar{p}) = \tilde{h} + \lambda \tilde{q}\bar{p} + \frac{\lambda^2 k}{4}\bar{p}^2 \,, \quad q(\bar{p}) = \tilde{q} + \frac{\lambda k}{2}\bar{p} \,, \tag{1.2}$$

where $\lambda$ is the deformation parameter. Our (more non-trivial) goal is to find a similar constraint that fixes the right-moving dependence of the correlators.

The other aspect concerns the method used to compute the correlation functions. Given the definition of the deformation in terms of an irrelevant flow, conformal perturbation theory comes in naturally, and this was the method used in [28] to analyse $J\bar{T}$ correlators, and in [29] (see also [30–32]) for the more involved $T\bar{T}$ case. However, note that the correlation functions computed with this method are UV divergent, and so need to be regulated and renormalized; however, it is *a priori* not clear whether any choice of UV regulator and which choices of counterterms would be allowed.

In this article, we approach the computation of $J\bar{T}$ correlators in a different way, which circumvents the issue of UV divergences and understanding what are the allowed counterterms.

Instead, we rely on the flow equation with respect to $\lambda$ satisfied by the energy eigenstates, the symmetry generators and an appropriately-defined set of operators to construct the correlation functions of interest. Our approach is similar in spirit to the one used in [33] to discuss $T\bar{T}$ correlators, though our basis of operators is different and the correlation functions we compute are manifestly finite throughout, as well as fully explicit.

More concretely, to fix the right-moving part of the correlator, we use the fact that $J\bar{T}$ - deformed CFTs were recently shown to possess an infinite-dimensional "pseudo-conformal" symmetry [34], implemented - at the classical level - by field-dependent generalizations of conformal and affine $U(1)$ transformations

$$v \to \bar{f}(v), \qquad v \equiv V - \lambda\phi, \tag{1.3}$$

where $V$ is the right-moving coordinate and $\phi$ is the bosonisation of the $U(1)$ current (with its zero mode removed, if working on the cylinder). The corresponding symmetry generators have been constructed in [12, 34] at the classical level, and in [11] at the full quantum level. The quantum construction relied on the existence of an alternate basis for these symmetries, denoted as the "flowed" representation, in which the generators - denoted with a $\sim$ - are simply defined to flow in the same way as the energy eigenstates. By construction, they satisfy a Virasoro-Kac-Moody algebra [35]; more non-trivially, they can also be shown to be conserved. A similar construction holds on the left-moving side. The generators of the left conformal and right pseudo-conformal (1.3) symmetries are given in terms of the flowed symmetry generators by

$$
\begin{aligned}
L_n &= \tilde{L}_n + \lambda H_R \tilde{J}_n + \frac{\lambda^2 k H_R^2}{4}\delta_{n,0}, & \bar{L}_n &= \tilde{\bar{L}}_n + \lambda : H_R\tilde{\bar{J}}_n : + \frac{\lambda^2 k H_R^2}{4}\delta_{n,0}, \\
J_n &= \tilde{J}_n + \frac{\lambda k}{2}H_R\delta_{n,0}, & \bar{J}_n &= \tilde{\bar{J}}_n + \frac{\lambda k}{2}H_R\delta_{n,0},
\end{aligned}
\tag{1.4}
$$

where the : : denote normal ordering, i.e. $\tilde{\bar{J}}_n$ is to the right of $H_R$ for $n > 0$, and to the left for $n < 0$. Note the above relation resembles a 'spectral flow' by the right-moving Hamiltonian, $H_R$. The algebra of these generators is Virasoro-Kac-Moody on the left and a non-linear deformation of it on the right; also, the left and right generators do not commute.

It is interesting to ask whether this deformed (Virasoro-Kac-Moody)$^2$ symmetry can help us fix the form of correlation functions, as it does in usual CFTs.

In two-dimensional CFTs, one can define primary operators - of dimension $(h, \bar{h})$ - either through their transformation properties under finite conformal transformations $z \to z'(z)$, $\bar{z} \to \bar{z}'(\bar{z})$

$$\mathcal{O}'(z', \bar{z}') = (\partial_z z')^{-h}(\partial_{\bar{z}}\bar{z}')^{-\bar{h}}\mathcal{O}(z, \bar{z}), \tag{1.5}$$

or through their commutation relations with the Virasoro generators

$$[L_n, \mathcal{O}(z)] = h(n+1)z^n\mathcal{O} + z^{n+1}\partial_z\mathcal{O}, \qquad n \geq -1 \tag{1.6}$$

and similarly on the right. For $n = \pm 1, 0$ this relation, together with the $SL(2, \mathbb{R})$ invariance of the vacuum, completely fixes the form of two and three-point functions, and highly constrains higher-point ones.

In $J\bar{T}$ - deformed CFTs, it is not clear how to define an analogue of (1.5). One may naively attempt to simply replace $\bar{z}$ in (1.5) by its field-dependent counterpart $\bar{z} - \lambda\phi$; however, this is in tension with the known fact that the left-moving dimensions depend on the right-moving momentum as in (1.2), which forces one to treat at least the right-movers in momentum space.

As we will show, it is nevertheless possible to come rather close to a definition of 'primary' operators in $J\bar{T}$ - deformed CFTs that is analogous to the momentum-space counterpart of (1.6). The correlation functions of the resulting operators are completely fixed in terms of the corresponding undeformed CFT correlators and, interestingly, take a form that highly resembles (1.1). To what extent our proposal is 'the' correct definition of primary operators in these non-local CFTs is left for future investigations.

This article is organised as follows. In section 2, we study a simple toy model for the $J\bar{T}$ deformation, which also exhibits two sets of symmetry generators related by an operator-dependent spectral flow. For this example we show, using only flow equations and the interplay of the symmetry generators, that general deformed primary correlators are completely determined by their undeformed counterparts. In section 3, we apply the same technique to compute general correlation functions in $J\bar{T}$ - deformed CFTs, making appropriate adjustments for the non-locality of the model. We conclude with a discussion in section 4. Various technical details of the symmetry algebras are detailed in the appendices, as well as a very explicit realisation of the toy model of section 2 in terms of deformed free bosons.

# 2   $J^1 \wedge J^2$ warmup

In this section, we would like to "warm up" for the construction of primary operators in $J\bar{T}$ - deformed CFTs by studying a simple toy model that also exhibits two possible bases for the symmetry generators, related by an operator-dependent spectral flow. This model is the so-called $J^1 \wedge J^2$ deformation of a two-dimensional CFT, which is the simplest possible Smirnov-Zamolodchikov deformation - built from two $U(1)$ currents - and also corresponds to a two-dimensional toy model for a four-dimensional gauge theory in presence of a $\theta$ term [36].

Since the $J^1 \wedge J^2$ deformation is exactly marginal, all the powerful tools of conformal symmetry are still applicable to the deformed theory; in particular, it is very clear what the primary operators are. Throughout this section, we will make an effort to treat this deformation in a language that is also applicable to the $J\bar{T}$ deformation, which will provide a very useful guiding principle for how to proceed when the conformal symmetry is deformed to the non-local, field-dependent symmetries (1.3).

## 2.1   Brief review of the $J^1 \wedge J^2$ deformation

We consider a one-parameter ($\lambda$) family of two-dimensional CFTs that posses two global $U(1)$ conserved currents, $J^{1,2}$. The actions describing the various members of this family are related via the flow equation

$$\frac{\partial S}{\partial \lambda} = -\int d^2x \, e^{\alpha\beta} \left( J^1_\alpha J^2_\beta \right)_\lambda \,, \tag{2.1}$$

where the bilinear operator appearing on the right-hand side is defined via point-splitting [19] and the current components are computed in the deformed CFT. The deformed spectrum has been understood in [36, 37], and certain aspects of correlation functions have been analysed in [29]. In this subsection, we will review some of these results, in a language that parallels the $J\bar{T}$ analysis of [11].

**Classical analysis**

It is useful to first understand the effect of the $J^1 \wedge J^2$ deformation at the classical level. This is perhaps simplest to present in Hamiltonian language. We thus consider a Hamiltonian density $\mathcal{H}(\pi_i, \phi_i)$, which admits at least two $U(1)$ symmetries that we associate to shifts in two scalars,

$\phi_{1,2}$. As a result, the Hamiltonian depends on these two canonical variables only through their spatial derivatives, $\phi'_{1,2}$. The two shift currents have components

$$J_t^a = \pi^a \sqrt{k}, \qquad J_\sigma^a = \partial_{\phi'_a} \mathcal{H} \sqrt{k}, \quad a = 1, 2, \tag{2.2}$$

where we have allowed for an arbitrary level[2], $k$. We will also consider the topologically conserved currents

$$\check{J}_t^a = \phi'^a \sqrt{k}, \qquad \check{J}_\sigma^a = \partial_{\pi_a} \mathcal{H} \sqrt{k}, \tag{2.3}$$

and will assume that in the undeformed CFT, with Hamiltonian density $\mathcal{H}^{(0)}(\pi_i, \phi_i)$, the combinations $J_\alpha^a \pm \check{J}_\alpha^a$ are (anti)chiral, which implies that

$$\partial_{\pi^a} \mathcal{H}^{(0)} = \pi^a, \qquad \partial_{\phi'_a} \mathcal{H}^{(0)} = \phi'_a. \tag{2.4}$$

The flow equation obeyed by the deformed Hamiltonian reads (in the convention $\epsilon_{t\sigma} = \epsilon^{\sigma t} = 1$)

$$\partial_\lambda \mathcal{H} = \epsilon^{\alpha\beta} J_\alpha^1 J_\beta^2 = k(\pi_2 \partial_{\phi'_1} \mathcal{H} - \pi_1 \partial_{\phi'_2} \mathcal{H}). \tag{2.5}$$

This equation can be solved by making the Ansatz

$$\mathcal{H}(\lambda) = \widetilde{\mathcal{H}}(\lambda) + \frac{\lambda^2 k^2}{2}(\pi_1^2 + \pi_2^2) + \lambda k(\phi'_1 \pi_2 - \phi'_2 \pi_1), \tag{2.6}$$

which implies that $\widetilde{\mathcal{H}}(\lambda)$ satisfies the equation

$$\partial_\lambda \tilde{\mathcal{H}} = k\pi_2(\partial_{\phi'_1} \tilde{\mathcal{H}} - \phi'_1) - k\pi_1(\partial_{\phi'_2} \tilde{\mathcal{H}} - \phi'_2). \tag{2.7}$$

This is solved by $\widetilde{\mathcal{H}}(\lambda) = \mathcal{H}^{(0)}$, using the initial condition (2.4). One can easily check, following e.g. [38], that the stress tensor computed from the resulting deformed Hamiltonian (2.6) is both symmetric and traceless, and thus the deformed theory remains a CFT. In this theory, it is useful we introduce the left/right Hamiltonian currents ($\mathcal{P}$ is the momentum density)

$$\mathcal{H}_{L,R} \equiv \frac{\mathcal{H} \pm \mathcal{P}}{2}, \qquad \mathcal{P} = \sum_i \pi_i \phi'_i. \tag{2.8}$$

We can now use the deformed Hamiltonian (2.6) and the definitions (2.2), (2.3) to compute the components of the deformed conserved currents. A basis for the currents that are now (anti)chiral is given by

$$\mathcal{J}_{L,R}^1 = \frac{\sqrt{k}}{2}(\pi_1 \pm \phi'_1 \pm \lambda k \pi_2), \qquad \mathcal{J}_{L,R}^2 = \frac{\sqrt{k}}{2}(\pi_2 \pm \phi'_2 \mp \lambda k \pi_1), \tag{2.9}$$

where the above expressions represent their time components and, by definition, $\mathcal{J}_{L,\sigma}^a = \mathcal{J}_{L,t}^a$, $\mathcal{J}_{R,\sigma}^a = -\mathcal{J}_{R,t}^a$. The Poisson brackets of the currents in this basis are diagonal and $\lambda$ - independent

$$\{\mathcal{J}_L^a(\sigma), \mathcal{J}_L^b(\tilde{\sigma})\} = -\{\mathcal{J}_R^a(\sigma), \mathcal{J}_R^b(\tilde{\sigma})\} = \frac{k}{2}\delta'(\sigma - \tilde{\sigma}), \qquad \{\mathcal{J}_L^a(\sigma), \mathcal{J}_R^b(\tilde{\sigma})\} = 0 \tag{2.10}$$

---

[2]Even if $k$ is an anomaly coeficient, for bosons it appears already at the level of the classical Poisson brackets. This allows one to understand many properties of the deformation with just a classical analysis.

and their commutators with the deformed Hamiltonian currents $\mathcal{H}_{L,R}(\lambda)$ yield the standard Witt-Kac-Moody algebra between (anti)chiral currents in a CFT. It is also interesting to note that the combinations

$$\hat{\mathcal{H}}_L \equiv \mathcal{H}_L - \frac{1}{k}\sum_a (\mathcal{J}_L^a)^2\,, \qquad \hat{\mathcal{H}}_R \equiv \mathcal{H}_R - \frac{1}{k}\sum_a (\mathcal{J}_R^a)^2\,, \tag{2.11}$$

are independent of $\lambda$. Thus, they equal the corresponding quantities in the undeformed CFT, which are nothing but the spectral-flow-invariant piece of the Hamiltonian currents. This structure is reminiscent of that of the left-moving Hamiltonian in $J\bar{T}$ - deformed CFTs [11].

One final observation that will be useful shortly is that, if we define the total currents $\mathcal{J}_\alpha^a \equiv \mathcal{J}_{L,\alpha}^a + \mathcal{J}_{R,\alpha}^a$, then the deforming operator can also be written in terms of them as

$$\mathcal{O}_{J^1 \wedge J^2} = \epsilon^{\alpha\beta} J_\alpha^1 J_\beta^2 = 2(\mathcal{J}_L^1 \mathcal{J}_R^2 - \mathcal{J}_R^1 \mathcal{J}_L^2) = \epsilon^{\alpha\beta} \mathcal{J}_\alpha^1 \mathcal{J}_\beta^2\,. \tag{2.12}$$

**Quantum analysis**

We now move on to the quantum theory, and place the $J^1 \wedge J^2$ - deformed CFT on a cylinder of radius $R$. Following [19], we consider eigenstates $|n_\lambda\rangle$ of the energy and the charge, whose shift and, respectively, winding charges are

$$n^a = \int d\sigma\, \langle J_t^a \rangle\,, \qquad w_a = \int d\sigma\, \langle \tilde{J}_t^a \rangle\,. \tag{2.13}$$

Since the deformation is integrable, it does not change the Hilbert space of states on the cylinder, but it induces a flow of the energy eigenstates, of the form

$$\partial_\lambda |n_\lambda\rangle = \mathcal{X}_{J\bar{J}} |n_\lambda\rangle \tag{2.14}$$

where $\mathcal{X}_{J\bar{J}} = -\mathcal{X}_{J\bar{J}}^\dagger$ is a well-defined operator acting on the Hilbert space, which we will determine shortly.

We would now like to understand how the energies and *chiral* charges $q^a, \bar{q}^a$ - i.e., the charges associated to the zero modes

$$\mathcal{Q}^a = \int d\sigma\, \mathcal{J}_L^a\,, \qquad \bar{\mathcal{Q}}^a = \int d\sigma\, \mathcal{J}_R^a \tag{2.15}$$

of the currents (2.9) - depend on $\lambda$. Using first order quantum-mechanical perturbation theory and the factorization properties of the Smirnov-Zamolodchikov operator in energy eigenstates, we find

$$\partial_\lambda E_n^\lambda = \langle n_\lambda | \partial_\lambda H | n_\lambda \rangle = R \left( \langle n_\lambda | J_\sigma^1 | n_\lambda \rangle \langle n_\lambda | J_t^2 | n_\lambda \rangle - \langle n_\lambda | J_t^1 | n_\lambda \rangle \langle n_\lambda | J_\sigma^2 | n_\lambda \rangle \right)$$
$$= \frac{1}{R} \left[ n_2(w_1 + \lambda k n_2) - n_1(w_2 - \lambda k n_1) \right]\,, \tag{2.16}$$

where $\partial_\lambda H$ is the spatial integral of (2.5) over the circle and we used (2.2) to compute the spatial components $J_\sigma^a$. Since the shift and winding charges defined above are quantized, and thus cannot flow with $\lambda$, this equation immediately integrates to the following expression for the deformed energies

$$E_n^\lambda = E_n^{(0)} + \frac{\lambda^2 k}{2R}(n_1^2 + n_2^2) + \frac{\lambda}{R}(w_1 n_2 - w_2 n_1)\,. \tag{2.17}$$

While this expression is entirely analogous to (2.6), note that it does not immediately follow from it.

The charges associated to the chiral conserved currents (2.9) are given by

$$q^a = \tilde{q}^a + \frac{\lambda k}{2}\epsilon^{ab}n_b\,, \qquad \bar{q}^a = \tilde{\bar{q}}^a - \frac{\lambda k}{2}\epsilon^{ab}n_b\,, \tag{2.18}$$

where $\tilde{q}_a, \tilde{\bar{q}}_a \equiv (n_a \pm w_a)/2$ stand for the undeformed (anti)chiral charges and $\epsilon_{12} = \epsilon^{12} = 1$. Using the state-operator correspondence to map the energies of eigenstates on the cylinder to the conformal dimensions of local operators on the plane, we find that the spectrum of left/right conformal dimensions in the $J^1 \wedge J^2$ - deformed CFT depends on $\lambda$ as

$$h = \tilde{h} + \lambda\epsilon_{ab}\tilde{q}^a n^b + \frac{\lambda^2 k}{4}\,n_a n^a\,, \qquad \bar{h} = \tilde{\bar{h}} - \lambda\epsilon_{ab}\tilde{\bar{q}}^a n^b + \frac{\lambda^2 k}{4}\,n_a n^a \tag{2.19}$$

where $\tilde{h}, \tilde{\bar{h}} = (E_n^{(0)} \pm P_n)R$ are the undeformed conformal dimensions.

Thus, the effect of the $J^1 \wedge J^2$ deformation on the spectrum of energies or, equivalently, on the local operator dimensions is precisely that of a simultaneous spectral flow in the two $U(1)$ directions, with charge-dependent parameters $\eta^a = -\bar{\eta}^a = \lambda\epsilon^{ab}n_b$ that are opposite on the left and the right. Note that the total momentum charge $n^a = q^a + \bar{q}^a$ is unaffected. One can easily check that $\hat{h} = h - (q_a q^a)/k$ is left invariant, as expected.

## 2.2 Flow of the states and of the symmetry generators

We would now like to determine the form of the operator $\mathcal{X}_{J\bar{J}}$ entering the flow equation (2.14) for the $J^a \wedge J^b$ - deformed energy eigenstates at least in the classical limit, in analogy with the results of [12] for $J\bar{T}$. Using first order quantum-mechanical perturbation theory and assuming the CFT degeneracies are dealt with, this operator can be read off from

$$\partial_\lambda|n\rangle_\lambda = \sum_{m \neq n} \frac{\langle m_\lambda|\partial_\lambda H|n_\lambda\rangle}{E_n^\lambda - E_m^\lambda}\,|m_\lambda\rangle\,, \tag{2.20}$$

where $\partial_\lambda H$ is the spatial integral of (2.5), performed on the $t = 0$ slice[3]

$$\partial_\lambda H = \int d\sigma\,\epsilon^{\alpha\beta}J_\alpha^1 J_\beta^2 = \int d\sigma\,\epsilon^{\alpha\beta}\mathcal{J}_\alpha^1\mathcal{J}_\beta^2\,, \tag{2.21}$$

where we used (2.12). To obtain a useful expression for $\mathcal{X}_{J\bar{J}}$, we follow the steps outlined in [33] for the case of $T\bar{T}$. This involves splitting the two current insertions in the deforming operator using a $\delta$ function, which is subsequently rewritten in terms of the Green's function on the cylinder, which satisfies

$$\partial_\sigma G(\sigma - \tilde{\sigma}) = \delta(\sigma - \tilde{\sigma}) - \frac{1}{R}\,. \tag{2.22}$$

After these manipulations, we obtain

$$\partial_\lambda H = \frac{2}{R}\epsilon_{ab}\mathcal{Q}^a\bar{\mathcal{Q}}^b - \partial_t\int d\sigma d\tilde{\sigma}\,G(\sigma - \tilde{\sigma})\mathcal{J}_t^1(\sigma)\mathcal{J}_t^2(\tilde{\sigma})\,, \tag{2.23}$$

where $\mathcal{Q}_a, \bar{\mathcal{Q}}_a$ are the chiral charge operators (2.15). The first term can also be written as a total time derivative by bosonising the currents $\mathcal{J}_{L,R}^a = \partial_\pm\varphi_{L,R}^a$ and noting that the zero modes of the chiral scalars thus introduced satisfy

$$[H, \varphi_{L,0}^a] = -i\mathcal{Q}^a\,, \qquad [H, \varphi_{R,0}^a] = -i\bar{\mathcal{Q}}^a\,. \tag{2.24}$$

---

[3]As explained in [12], for $t \neq 0$ the flow operator will receive additional contributions proportional to $\partial_\lambda E_n^\lambda$.

Then, $2\epsilon_{ab}\mathcal{Q}^a\bar{\mathcal{Q}}^b = [H, i\epsilon_{ab}(\varphi^a_{L,0}\bar{\mathcal{Q}}^b + \mathcal{Q}^a\varphi^b_{R,0})] = -i\frac{d}{dt}(\ldots)$. The next step is to use an integral representation of the denominator in (2.20) to rewrite it as

$$\partial_\lambda |n_\lambda\rangle = -i \sum_{m \neq n} \int_{-\infty}^{0} dt\, e^{t\epsilon} |m_\lambda\rangle \langle m_\lambda | \partial_\lambda H(t) | n_\lambda\rangle, \tag{2.25}$$

where $\epsilon > 0$ is an infinitesimal regulator. Performing the integral and taking $\epsilon \to 0$, we find

$$\partial_\lambda |n_\lambda\rangle = -i \sum_{m \neq n} |m_\lambda\rangle \langle m_\lambda \left| \frac{\epsilon_{ab}}{R}\left(\varphi^a_{L,0}\bar{\mathcal{Q}}^b + \mathcal{Q}^a\varphi^b_{R,0}\right) - \int d\sigma d\breve{\sigma}\, G(\sigma - \breve{\sigma})\mathcal{J}^1_t(\sigma)\mathcal{J}^2_t(\breve{\sigma}) \right| n_\lambda\rangle. \tag{2.26}$$

It is easy to argue [12] that the matrix elements of the first term will vanish between different eigenstates, and thus will drop from the sum[4]. It is also easy to see, e.g. using a Fourier decomposition, that the second term will only have non-zero matrix elements if the eigenstates are different. Then, in the classical limit, the flow 'operator' for the energy eigenstates is simply given by

$$\mathcal{X}_{J\bar{J}} = i \int d\sigma d\breve{\sigma}\, G(\sigma - \breve{\sigma})\mathcal{J}^1_t(\sigma)\mathcal{J}^2_t(\breve{\sigma}). \tag{2.27}$$

We would now like to derive how the various currents flow with respect to $\lambda$. For our purposes, it will be sufficient to understand this at the classical level. If a classical current is left invariant by

$$\tilde{\mathcal{D}}_\lambda \equiv \partial_\lambda - i\{\mathcal{X}_{J\bar{J}}, \cdot\}, \tag{2.28}$$

then we will assume this implies that at the quantum level, it will flow in the same way, (2.14), as the energy eigenstates. Introducing the total momentum operator

$$\Pi^a = \mathcal{Q}^a + \bar{\mathcal{Q}}^a = \sqrt{k} \int d\sigma\, \pi^a \tag{2.29}$$

we find that the various currents satisfy

$$\tilde{\mathcal{D}}_\lambda \mathcal{J}^a_L = \frac{k}{2R}\epsilon^{ab}\Pi_b, \quad \tilde{\mathcal{D}}_\lambda \mathcal{J}^a_R = -\frac{k}{2R}\epsilon^{ab}\Pi_b, \quad \tilde{\mathcal{D}}_\lambda \mathcal{H}_L = \frac{1}{R}\epsilon_{ab}\mathcal{J}^a_L\Pi^b, \quad \tilde{\mathcal{D}}_\lambda \mathcal{H}_R = -\frac{1}{R}\epsilon_{ab}\mathcal{J}^a_R\Pi^b. \tag{2.30}$$

Consequently, the following combinations

$$\tilde{\mathcal{H}}_L \equiv \mathcal{H}_L - \frac{\lambda}{R}\epsilon_{ab}\mathcal{J}^a_L\Pi^b + \frac{\lambda^2 k}{4R^2}(\Pi_a)^2, \quad \tilde{\mathcal{J}}^a_L \equiv \mathcal{J}^a_L - \frac{\lambda k}{2R}\epsilon_{ab}\Pi^b, \tag{2.31}$$

$$\tilde{\mathcal{H}}_R \equiv \mathcal{H}_R + \frac{\lambda}{R}\epsilon_{ab}\mathcal{J}^a_R\Pi^b + \frac{\lambda^2 k}{4R^2}(\Pi_a)^2, \quad \tilde{\mathcal{J}}^a_R \equiv \mathcal{J}^a_R + \frac{\lambda k}{2R}\epsilon_{ab}\Pi^b, \tag{2.32}$$

flow in the same way as the energy eigenstates. In terms of the (dimensionless) Fourier modes of these generators, now seen as operators, we have

$$\tilde{J}^a_m = J^a_m - \frac{k\eta^a}{2}\delta_{m,0}, \quad \tilde{L}_m = L_m - \eta_a J^a_m + \frac{k\eta_a\eta^a}{4}\delta_{m,0},$$

$$\tilde{\bar{J}}^a_m = \bar{J}^a_m - \frac{k\bar{\eta}^a}{2}\delta_{m,0}, \quad \tilde{\bar{L}}_m = \bar{L}_m - \bar{\eta}_a\bar{J}^a_m + \frac{k\eta_a\eta^a}{4}\delta_{m,0}, \tag{2.33}$$

---

[4]Note this derivation of the flow operator is significantly easier than its $J\bar{T}$ [12] and $T\bar{T}$ (currently not understood) counterpart, where the main difficulty lies is finding the projection of each of the two terms on the energy eigenstates.

where we introduced the operator-dependent spectral flow parameter

$$\eta_a = \lambda \epsilon_{ab} \Pi^b = -\bar\eta_a \, . \tag{2.34}$$

Thus, we find that in $J^1 \wedge J^2$ - deformed CFTs, there exist two interesting bases for the symmetry generators, which are related by an operator-dependent spectral flow. One basis consists of the generators $L_m, J_m, \bar{L}_m$ and $\bar{J}_m$, which directly implement conformal and affine $U(1)$ transformations. The other basis consists of the generators $\tilde{L}_m, \tilde{J}_m$ and their right-moving counterparts, which have the property that they flow with $\lambda$ in the same way as the energy eigenstates, namely

$$|n_\lambda\rangle = U_\lambda |n_0\rangle \, , \qquad \tilde{L}_m(\lambda) = U_\lambda L_m(0) U_\lambda^{-1} \, , \tag{2.35}$$

where $U_\lambda = \mathcal{P} e^{\int \mathcal{X}_{J\bar{J}} \, d\lambda}$ and $|n_0\rangle, L_m(0)$ are the energy eigenstates and, respectively, the symmetry generators in the undeformed CFT. This structure exactly parallels the one observed for $J\bar{T}$ -deformed CFTs [11].

In either basis, the symmetry algebra consists of two commuting copies of the Virasoro-Kac-Moody algebra, consistently with the fact that the spectral flow parameter, even though operator-valued, commutes with all the modes of the symmetry currents. The Hilbert space is organised into highest-weight representations of this algebra, which can be built with respect to either $L_m$ or $\tilde{L}_m$. Note that primary states $|h_\lambda\rangle$ with respect to one basis will also be primary with respect to the other; however, the descendants in one basis will generally be a linear combination of descendants of the same level in the other. Note also that, due to (2.35), the eigenvalues of the zero modes of the flowed generators are independent of $\lambda$, and thus will equal those of the undeformed CFT

$$\tilde{L}_0|n_\lambda\rangle = \tilde{h}|n_\lambda\rangle \, , \quad \tilde{J}_0|n_\lambda\rangle = \tilde{q}|n_\lambda\rangle \, , \qquad \tilde{\bar{L}}_0|n_\lambda\rangle = \tilde{\bar{h}}|n_\lambda\rangle \, , \quad \tilde{\bar{J}}_0|n_\lambda\rangle = \tilde{\bar{q}}|n_\lambda\rangle \, . \tag{2.36}$$

The flow (2.19), (2.18) of the conformal dimensions and chiral charges is then explained by the relation (2.33) between the flowed and the standard conformal generators, where the operator-dependent spectral flow parameter takes on its eigenvalue corresponding to the state under consideration.

## 2.3  From states and generators of symmetries to operators

We would now like to compute correlation functions of primary operators in the $J^1 \wedge J^2$ - deformed CFT. Of course, since the deformed theory is still a CFT and the conformal dimensions of primary operators are known (2.19), one can immediately write down the primary two- and three-point functions in this theory up to an overall normalization. In this section, we will show that it is in fact possible to determine *all* the correlation functions in this model exactly in terms of the correlation functions of the undeformed CFT. That this should have been possible is implied by the results of [29] on the flow of correlation functions in $J^1 \wedge J^2$ - deformed CFTs; our method allows, in addition, to write down an entirely explicit expression for the relation between the deformed and undeformed correlators of primary operators.

Since this exercise is supposed to serve as warm-up for the more difficult $J\bar{T}$ case, we would like to phrase our computations entirely in terms of states and symmetry generators on the cylinder, which are quantities that we have access to also in $J\bar{T}$ - deformed CFTs. In particular, we will do our best to avoid resorting to radial quantization or the state-operator correspondence, which have not (yet) been formulated for these theories. The plan of this section is to slowly build some intuition for our construction; for the actual proposal, the reader can skip to (2.48).

An observable that can be straightforwardly constructed from the above building blocks is the cylinder two-point function, seen as the overlap of an in- and an out-state created by acting with a (primary) operator on the vacuum

$$\mathcal{O}(w)|0\rangle = e^{wh}e^{e^{w}L_{-1}}|h\rangle. \tag{2.37}$$

In the above, $|h\rangle$ is a primary state on the cylinder, $w = \tau + i\sigma$ is the complex coordinate on the cylinder ($\tau = it$) and $L_{\pm 1} = -e^{\pm w}\partial_w$, $L_0 = -\partial_w$ are the global conformal generators on the cylinder, which satisfy the $SL(2, \mathbb{R})$ algebra with the usual conventions. There is a completely analogous contribution from the right-moving side, which we do not write to avoid cluttering.

The equation above is derived in several steps: first, one uses the state-operator correspondence to map the primary state on the cylinder to a primary operator inserted at the origin of the plane $|h\rangle \to \mathcal{O}_{pl}(0)|0\rangle$. This may be understood as the definition of the primary operator. Next, one can define an operator at an arbitrary location $z$ on the plane by translating[5] it with $L_{-1}^{pl} = -\partial_z$, i.e. $\mathcal{O}_{pl}(z) = e^{zL_{-1}^{pl}}\mathcal{O}_{pl}(0)e^{-zL_{-1}^{pl}}$. When acting on the vacuum, which is annihilated by the right $L_{-1}$ factor, we obtain the plane analogue of (2.37), $\mathcal{O}_{pl}(z)|0\rangle = e^{zL_{-1}^{pl}}|h\rangle$. The final step is to map the resulting expression back to the cylinder via $z = e^w$, using the fact that in radial quantization, $L_{-1}^{pl}$ is identified with its counterpart on the cylinder, as well as the relation $\mathcal{O}_{(cyl)}(w) = e^{wh}\mathcal{O}_{pl}(z)$, which follows from the transformation properties (1.5) of primary operators under conformal transformations. Of course, almost none of these steps would hold[6] in $J\bar{T}$ - deformed CFTs, but the final result is a well-defined expression on the cylinder, which we could simply use it to *define* the operators that we would like to consider.

It may in fact be possible to give an interpretation to (2.37) directly on the cylinder, by thinking of the primary state as being created by an operator insertion at $\tau = -\infty$, i.e. $|h\rangle = \lim_{\tau \to -\infty} e^{-h\tau}\mathcal{O}(\tau)$, and of the exponentiated $L_{-1}$ as implementing a conformal transformation that brings the point at $-\infty$ to finite distance. Again, one needs to be careful about the fact that $L_{-1}$ is not a Hermitean operator; however, as we show in appendix A, its action on a primary state can be reproduced by the action of a combination of the Hermitean operators $L_1 + L_{-1}$ and $i(L_1 - L_{-1})$, with appropriately chosen coefficients. While this picture does help avoid the map to radial quantization on the plane when constructing the action of these operators, it does not necessarily help justify a definition of the form (2.37) for $J\bar{T}$ - deformed CFTs[7].

---

[5]Note that this is not a unitarily-implemented translation, even though the prefactor does take the familiar form

$$zL_{-1} + \bar{z}\bar{L}_{-1} = -(z\partial_z + \bar{z}\partial_{\bar{z}}) = -(t\partial_t + x\partial_x) = iHt - iPx \quad \text{with} \quad H = i\partial_t, P = -i\partial_x, z = x + it,$$

because $L_{-1}$ is not Hermitean in radial quantization, and thus $H, P$ are not, either. In fact, a translation is not a symmetry of the CFT in radial quantization, because the latter singles out a special point - the origin of the plane - where operators are inserted. The fact that $\mathcal{O}_{pl}(z)$ takes the form quoted in the text is implied by the Ward identities (1.6) associated with translations, which are independent of the quantization we choose [39].

[6]Some of the complications that one encounters are: i) The map from the cylinder to the plane, assuming it can be well-defined, will be field-dependent (1.3), and thus $\tau \to -\infty$ and $\tau = 0$ on the cylinder will not map to a fixed location and, respectively, a fixed circle on the plane. Relatedly, dilatations correspond to a field-dependent symmetry in $J\bar{T}$. This makes it difficult to formulate a state-operator correspondence precisely, even if intuitively such a map may exist; ii) The generator of right-moving translations on the plane - which are standard symmetries - does not appear to be identified with $\bar{L}_{-1}$ on the cylinder, which implements a field-dependent transformation. This can be easily established by noting that $L_{-1}$ and $\bar{L}_{-1}$ on the cylinder do not commute (C.6) (except when $R \to \infty$), whereas they obviously do on the plane; iii) Instead of mapping back to the cylinder, one could simply attempt to compute correlation functions on the plane. However, in this case it is not clear how to define Hermitean conjugation, given the general lack of understanding of radial quantization in this theory. In particular, since $L_1^{pl}$ implements a field-dependent symmetry, it is not clear whether the coordinate appearing in the out bra should rather be a field-dependent coordinate. Using a different quantization, such as [40], does not appear to help, either.

[7]The reason is that in $J\bar{T}$ - deformed CFTs, the transformation taking the point at $-\infty$ to finite distance is field-

To compute correlators, we will also need the expression for the out state

$$\langle 0|\mathcal{O}(w) = e^{-wh}\langle h|e^{e^{-w}L_1}, \tag{2.38}$$

which follows from the simple fact that on the cylinder, hermitean conjugation[8] sends $w = i(t + \sigma) \to -w$. Taking the overlap, one obtains

$$\langle \mathcal{O}(w_1)\mathcal{O}(w_2)\rangle = e^{-hw_{12}}\langle h|e^{e^{-w_1}L_1}e^{e^{w_2}L_{-1}}|h\rangle = e^{-hw_{12}}e^{-2h\ln(1-e^{-w_{12}})} = \left(2\sinh\frac{w_{12}}{2}\right)^{-2h}, \tag{2.39}$$

where we used the relation (A.5) and the primary condition. This is of course the correct result on the cylinder, where the dimension is given by (2.19).

As advertised, a nice feature of this method is that it recasts the computation of the cylinder two-point function only in terms of states and symmetry generators, which are in principle also accessible in $J\bar{T}$ - deformed CFTs[9]. On the down side, this method is limited to two-point functions only. Also, it is not clear whether (2.37) provides a satisfactory definition for the primary operators in $J\bar{T}$ - deformed CFTs, since we were unable to motivate this particular choice for the action of the operator.

To proceed, it is useful to perform the calculation of the two-point function via overlaps in a slightly different way, which explicitly involves the flowed generators (2.33). In terms of them, the two-point function (2.39) reads

$$\langle \mathcal{O}(w_1)\mathcal{O}(w_2)\rangle = e^{-hw_{12}}\langle h_\lambda|e^{e^{-w_1}L_1}e^{e^{w_2}L_{-1}}|h_\lambda\rangle = e^{-hw_{12}}\langle h_\lambda|e^{e^{-w_1}(\tilde{L}_1+\eta^a\tilde{J}_1^a)}e^{e^{w_2}(\tilde{L}_{-1}+\eta^a\tilde{J}_{-1}^a)}|h_\lambda\rangle, \tag{2.40}$$

where $\eta^a = \lambda\epsilon^{ab}\Pi_b$ is the spectral flow operator (2.34) and we have reinstated the label $\lambda$ on the state, to emphasize its flow properties. Since $\eta^a$ commutes with all the modes of the currents and, inside this correlator, it is acting on the state $|h_\lambda\rangle$, then we can simply replace it by its eigenvalue $\eta_{\mathcal{O}}^a = \lambda\epsilon^{ab}n_b$ in this state. We then observe that the states and all the operators in the above expression flow with $\lambda$ in exactly the same way (2.35), and thus this correlator will be identical to the corresponding one in the undeformed CFT. In particular, its $\lambda$ dependence is entirely due to the explicit $\lambda$ - dependence of $\eta_{\mathcal{O}}^a$. Rather than evaluating this correlator in the undeformed CFT, we will prefer to work with the flowed states and generators in the deformed theory, keeping in mind that the two computations are simply related by conjugation by the unitary operator $U_\lambda$, defined in (2.35).

The correlator can now be evaluated using the following BCH-type formula

$$e^{e^w(\tilde{L}_{-1}+\eta\tilde{J}_{-1})}e^{-e^w\tilde{L}_{-1}} = e^{\eta\sum_{n=1}^\infty \frac{1}{n}e^{nw}\tilde{J}_{-n}} \tag{2.41}$$

---

dependent, and thus one may wonder whether one should replace the field-independent label $e^w$ in the definition (2.37) by a field-dependent one. In fact, one can easily check that an operator defined via (2.37) in $J\bar{T}$ - deformed CFTs lacks a number of desirable properties - for example, the action of the field-independent operator $e^{\alpha L_0}$ does not correspond to the translation $w \to w + \alpha$ in the label of the operator, as expected, except in the $R \to \infty$ limit. For more details, see appendix C.

[8]In radial quantization on the plane, one has instead $\langle 0|\mathcal{O}(z) \equiv (e^{z'L_{-1}}|h\rangle)^\dagger = z^{-2h}\langle h|e^{L_1/z}$, using $z' = 1/z$, which follows from the action of hermitean conjugation on the cylinder and the map $z = e^w$.

[9]It is clear that the overlap of two states of the form (2.37) and (2.38) can be evaluated also in in $J\bar{T}$ - deformed CFTs, since algebra of the unflowed generators is known; see e.g. appendix C. If the operators in question only depend on the left-moving coordinate, then only the commutation relations of the left-moving generators, $L_{\pm 1}$, are relevant. These are simply $SL(2,\mathbb{R})$ commutation relations, and one can proceed exactly as above to find the deformed left-moving piece of the two-point function, which has the expected form. We are however interested in its behaviour on the right-moving side. There, one encounters the complication that the algebra of $\bar{L}_{\pm 1,0}$ does not close, but instead generates the entire Kac-Moody tower. While the calculation is still in principle doable, we will see in the sequel that this proposal is in fact *not* equivalent to the one we finally settle for in the case of $J\bar{T}$ - deformed CFTs.

derived in appendix A, which also holds if we sum over several currents. Using this, the overlap is

$$\langle \mathcal{O}(w_1)\mathcal{O}(w_2)\rangle = e^{-hw_{12}}\langle h_\lambda|e^{e^{-w_1}\tilde{L}_1}e^{\eta_{\mathcal{O}}\sum_{n=1}^{\infty}\frac{1}{n}e^{-nw_1}\tilde{J}_n}e^{\eta_{\mathcal{O}}\sum_{n=1}^{\infty}\frac{1}{n}e^{nw_2}\tilde{J}_{-n}}e^{e^{w_2}\tilde{L}_{-1}}|h_\lambda\rangle \tag{2.42}$$

$$= e^{-(h-\tilde{h})w_{12}}e^{\frac{k\eta_{\mathcal{O}}^2}{2}\sum_{n=1}^{\infty}\frac{1}{n}e^{-nw_{12}}}\langle\tilde{\mathcal{O}}(w_1)e^{\eta_{\mathcal{O}}\sum_{n=1}^{\infty}\frac{1}{n}e^{nw_2}\tilde{J}_{-n}}e^{\eta_{\mathcal{O}}\sum_{n=1}^{\infty}\frac{1}{n}e^{-nw_1}\tilde{J}_n}\tilde{\mathcal{O}}(w_2)\rangle,$$

where in the first line we have used the fact that hermitean conjugation sends $w \to -w$, in addition to its usual effect on the generators, and in the second line we used the BCH identity $e^A e^B = e^B e^A e^{[A,B]}$, valid if $[A,B] \propto I$, to commute the exponentials of the currents. We additionally modeled the action of $e^{e^w\tilde{L}_{-1}}$ on the state $|h_\lambda\rangle$ by the action of an auxiliary operator $\tilde{\mathcal{O}}(w)$, acting on the vacuum

$$\tilde{\mathcal{O}}(w)|0_\lambda\rangle = e^{w\tilde{h}}e^{e^w\tilde{L}_{-1}}|h_\lambda\rangle. \tag{2.43}$$

This relation follows from the corresponding relation in the undeformed CFT, by conjugation with $U_\lambda$. The operator $\tilde{\mathcal{O}}$ is simply *defined* via the relation

$$\tilde{\mathcal{O}}(w) \equiv U_\lambda \mathcal{O}_{CFT}(w)U_\lambda^{-1}, \tag{2.44}$$

and need not correspond to a physical operator in the deformed CFT. Given this definition, it follows that $\tilde{\mathcal{O}}(w)$ satisfies the same Ward identities with the flowed currents as the corresponding quantities in the undeformed CFT, namely

$$[\tilde{J}_n^a, \tilde{\mathcal{O}}(w)] = \tilde{q}^a\, e^{nw}\tilde{\mathcal{O}}(w), \qquad n \geq 0. \tag{2.45}$$

This in turn implies that, for any coefficients $\alpha_n$,

$$e^{\alpha_n\tilde{J}_n^a}\tilde{\mathcal{O}}(w)e^{-\alpha_n\tilde{J}_n^a} = e^{\alpha_n\tilde{q}^a e^{nw}}\tilde{\mathcal{O}}(w), \qquad e^{-\alpha_n\tilde{J}_{-n}^a}\tilde{\mathcal{O}}(w)e^{\alpha_n\tilde{J}_{-n}^a} = e^{-\alpha_n\tilde{q}^a e^{-nw}}\tilde{\mathcal{O}}(w), \tag{2.46}$$

where the second relation is the hermitean conjugate of the first (using $\mathcal{O}_q^\dagger(w) = \mathcal{O}_{-q}(-w)$). Using these relations to commute the $\tilde{J}_n$ and $\tilde{J}_{-n}$ factors past the adjacent operators (and noting the one on the left has charge $-\tilde{q}$), the final answer that we obtain for the correlator is

$$\langle \mathcal{O}(w_1)\mathcal{O}(w_2)\rangle = e^{w_{12}(\tilde{h}-h)}e^{\frac{k\eta_{\mathcal{O}}^2}{2}\sum_{n=1}^{\infty}\frac{1}{n}e^{-nw_{12}}+2\eta_{\mathcal{O}}\tilde{q}\sum_{n=1}^{\infty}\frac{1}{n}e^{-nw_{12}}}\langle\tilde{\mathcal{O}}(w_1)\tilde{\mathcal{O}}(w_2)\rangle$$

$$= \frac{e^{-hw_{12}}}{(1-e^{-w_{12}})^{2(\tilde{h}+\eta_{\mathcal{O}}\tilde{q}+\frac{k}{4}\eta_{\mathcal{O}}^2)}}, \tag{2.47}$$

where we used $\sum_{n=1}^{\infty}\frac{1}{n}e^{-nw} = -\ln(1-e^{-w})$ and the fact that the $\tilde{\mathcal{O}}$ two-point function is identical to the one in the undeformed CFT, where the operator dimension was $\tilde{h}$. Thus, this method precisely reproduces the shift of the operator dimensions due to the $J^1 \wedge J^2$ deformation inside the correlation function.

We are now ready to present our general construction. We define a set of "operators" $\tilde{\mathcal{O}}(w)$ as solutions to the flow equation

$$\partial_\lambda\tilde{\mathcal{O}}(w) = [\mathcal{X}_{J\bar{J}}, \tilde{\mathcal{O}}(w)], \tag{2.48}$$

with the initial condition that $\tilde{\mathcal{O}}(w)_{\lambda=0}$ equal the CFT primary operators inserted at a point $w$ on the cylinder. One should think of these operators as being defined on the $t = 0$ slice[10], despite the $w$ label (which has no physical meaning, except at $\lambda = 0$). This flow equation can

---

[10]For the original CFT operators, we should therefore write $\mathcal{O}(w) = e^{wL_0}\mathcal{O}(0)e^{-wL_0}$.

certainly be integrated to an equation of the form (2.44), though it will not in general produce a local operator in the deformed CFT (see appendix B for an explicit example). Nevertheless, the correlation functions[11] of the flowed operators will be identical to those in the undeformed CFT, by virtue of the fact that they obey the same flow equation as the deformed states. Such operators have been previously considered in [33]. One of their nice features is that they can clearly also be defined in $J\bar{T}$ - deformed CFTs.

Our task is now to relate the correlation functions of the physical primary operators, $\mathcal{O}(w)$, in the deformed CFT to those of the unphysical operators $\tilde{\mathcal{O}}(w)$, which simply equal the original CFT correlators. For this, we need to relate $\mathcal{O}(w)$ and $\tilde{\mathcal{O}}(w)$ in the deformed theory. This is straightforward in $J^1 \wedge J^2$ - deformed CFTs, which are conformally invariant, and so primary operators must obey the usual conformal Ward identity[12]

$$= e^{nw}[\, nh\mathcal{O}(w) + \partial_w\mathcal{O}(w)], \qquad n \geq -1,$$
$$[J_n^a, \mathcal{O}(w)] = q^a e^{nw}\mathcal{O}(w), \qquad n \geq 0,$$

where, according to (2.19), (2.18)

$$h = \tilde{h} + \eta_{\mathcal{O}}^a \tilde{q}_a + \frac{k\eta_{\mathcal{O}}^2}{4}, \qquad q^a = \tilde{q}^a + \frac{k\eta_{\mathcal{O}}^a}{2}, \qquad \eta_{\mathcal{O}}^a = \lambda \epsilon^{ab} n_b \qquad (2.49)$$

and similarly for the right-movers.

On the other hand, the flow equation applied to the original Ward identity implies that

$$[\tilde{L}_n, \tilde{\mathcal{O}}(w)] = e^{nw}[\, n\tilde{h}\tilde{\mathcal{O}}(w) + \partial_w\tilde{\mathcal{O}}(w)], \qquad n \geq -1, \qquad (2.50)$$

together with (2.45). Using the relationship (2.33) between the flowed and the unflowed generators, it is easy to show that the relation between $\mathcal{O}(w,\bar{w})$ and $\tilde{\mathcal{O}}(w,\bar{w})$ is given by

$$\mathcal{O}(w,\bar{w}) = e^{A_{\mathcal{O}}w + B_{\mathcal{O}}\bar{w}} e^{\eta_{\mathcal{O}}^a \sum_{n=1}^\infty \frac{1}{n} e^{nw}\tilde{J}_{-n}^a + \bar{\eta}_{\mathcal{O}}^a \sum_{n=1}^\infty \frac{1}{n} e^{n\bar{w}}\tilde{\bar{J}}_{-n}^a} \tilde{\mathcal{O}}(w,\bar{w}) e^{-\eta_{\mathcal{O}}^a \sum_{n=1}^\infty \frac{1}{n} e^{-nw}\tilde{J}_n^a - \bar{\eta}_{\mathcal{O}}^a \sum_{n=1}^\infty \frac{1}{n} e^{-n\bar{w}}\tilde{\bar{J}}_n^a},$$
$$(2.51)$$

where

$$A_{\mathcal{O}} = \eta_a q^a + \eta_{\mathcal{O}}^a \tilde{J}_0^a - \frac{k}{4}(\eta_{\mathcal{O}}^a)^2 - \eta_{\mathcal{O}}^a \tilde{q}^a, \qquad B_{\mathcal{O}} = \bar{\eta}^a \bar{q}_a + \bar{\eta}_{\mathcal{O}}^a \tilde{\bar{J}}_0^a - \frac{k}{4}(\bar{\eta}_{\mathcal{O}}^a)^2 - \bar{\eta}_{\mathcal{O}}^a \tilde{\bar{q}}^a \qquad (2.52)$$

and we have reinstated the right-movers. The subscript on the operators $A_{\mathcal{O}}, B_{\mathcal{O}}$ indicate that they depend on the charges of the particular operator under consideration. One should also be careful to distinguish the operators $\eta^a$ from their eigenvalue $\eta_{\mathcal{O}}^a$, with $[\eta^a, \mathcal{O}(w)] = \eta_{\mathcal{O}}^a \mathcal{O}(w)$. Of course, in the case at hand we have $\bar{\eta}_a = -\eta_a$, since the spectral flow (2.33) acts in opposite ways on the left- and the right-movers.

One can easily check that $\mathcal{O}(w)$ satisfies the usual Hermiticity condition $\mathcal{O}_q^\dagger(-w) = \mathcal{O}_{-q}(w)$

$$\mathcal{O}_q^\dagger(-w) = e^{-\eta_{\mathcal{O}}^a \sum \frac{1}{n} e^{nw} J_{-n}} \tilde{\mathcal{O}}_q^\dagger(-w) e^{\eta_{\mathcal{O}}^a \sum \frac{1}{n} e^{-nw} J_n^a - A_{\mathcal{O}}^\dagger w} = e^{-A_{\mathcal{O}}^\dagger w} \mathcal{O}_{-q}(w) e^{-A_{\mathcal{O}}^\dagger w} = \mathcal{O}_{-q}(w), \quad (2.53)$$

where we used the fact that the charges of $\mathcal{O}^\dagger$ are opposite from those of $\mathcal{O}$ and that $A_{\mathcal{O}}$ is hermitean. We have again dropped the right-movers, for simplicity.

Thus, we find a rather simple, closed-form relation between the primary operators of interest and the auxiliary operators $\tilde{\mathcal{O}}$ that we defined through the flow equation. In appendix B,

---

[11] Even if our notation will be mostly euclidean, we will tacitly consider the analytic continuation to Wightman functions, in terms of which the flow picture makes sense.

[12] This follows from the usual relation (1.6) on the plane, using $z = e^w$ and $\mathcal{O}_{pl}(z) = e^{-wh}\mathcal{O}_{cyl}(w)$.

we present explicit expressions for both sets of operators for the case of $J^1 \wedge J^2$ - deformed free bosons, which make it clear that $\tilde{\mathcal{O}}(w)$ are non-local, and thus do not correspond to physical operators that we would like to consider otherwise.

In view of our previous discussion, note that the left prefactor in the relation between $\mathcal{O}$ and $\tilde{\mathcal{O}}$ above can be understood by acting with both sides of equation (2.51) on the vacuum, case in which it can be mapped to the relation (2.41) between $L_{-1}$ and $\tilde{L}_{-1}$. Heuristically, if one thinks of the primary state as being created by the insertion of a primary operator at $\tau = -\infty$ on the cylinder, then the action of $\mathcal{O}(w)$ can be obtained by conformally mapping the point at infinity to finite distance using the standard conformal generator $L_{-1}$, whereas the action of $\tilde{\mathcal{O}}(w)$ is obtained by using instead the flowed generator $\tilde{L}_{-1}$. However, this intuition does not help in understanding the right-hand factor in (2.51), nor why is the spectral flow operator evaluated to $\eta_{\mathcal{O}}$, even when not acting with (2.51) on the vacuum, from either the left or the right. Therefore, while intuitively useful and correct in the particular computation of the two-point function above, the state overlap picture fails to identify the general map between the two operators[13]. In the following subsection, we use the relation (2.51) to compute arbitrary correlators in the $J^1 \wedge J^2$ - deformed CFT.

## 2.4 Correlation functions and a bootstrap check

Given the expression (2.51) for the primary operators in the $J^1 \wedge J^2$ - deformed CFT in terms of the auxiliary operators $\tilde{\mathcal{O}}(w)$, whose correlation functions are known, computing correlation functions of $\mathcal{O}(w)$ becomes simply a matter of properly commuting the current modes through.

Let us start with the two-point function. Evaluating two copies of (2.51) in the vacuum, we find

$$
\begin{aligned}
\langle \mathcal{O}(w_1) O(w_2) \rangle &= e^{-\sum_{i=1}^{2}(\frac{k}{4}\eta_i^2 + \eta_i^a \tilde{q}_i^a)w_i} \\
&\times \langle e^{(\eta_a q_1^a + \eta_1^a \tilde{J}_0^a)w_1} \tilde{\mathcal{O}}(w_1) e^{-\eta_1^a \sum_{n=1}^{\infty} \frac{1}{n} e^{-nw_1} \tilde{J}_n^a} e^{(\eta_a q_2^a + \eta_2^a \tilde{J}_0^a)w_2} e^{\eta_2^a \sum_{n=1}^{\infty} \frac{1}{n} e^{nw_2} \tilde{J}_{-n}^a} \tilde{\mathcal{O}}(w_2) \rangle ,
\end{aligned}
\tag{2.54}
$$

where we have used the fact that the vacuum is annihilated from the left by $\tilde{J}_n^a$ with $n > 0$ and from the right by $\tilde{J}_{-n}^a$. Next, we note that in the above, the leftmost $\eta_a$ and $\tilde{J}_0^a$ will evaluate to zero, since they are acting on the vacuum; as for the middle ones, they are evaluated on the eigenstate created by $\tilde{\mathcal{O}}(w_2)|0\rangle$, so they evaluate to $\eta_2^a$ and, respectively, $\tilde{q}_2^a$. The remaining manipulations are identical to those performed in the previous section, and we obtain[14]

$$
\langle \mathcal{O}(w_1) O(w_2) \rangle = e^{-\sum_{i=1}^{2}(\frac{k}{4}\eta_i^2 + \eta_i^a \tilde{q}_i^a)w_i + \eta_2^a(q_2^a + \tilde{q}_2^a)w_2} e^{-(\frac{k}{2}\eta_1^a \eta_2^a + \eta_1^a \tilde{q}_2^a + \eta_2^a \tilde{q}_1^a)\sum_n \frac{1}{n} e^{-nw_{12}}} \langle \tilde{\mathcal{O}}(w_1) \tilde{O}(w_2) \rangle .
\tag{2.55}
$$

Performing the sum and using charge conservation, which sets $-\eta_1 = \eta_2 \equiv \eta_{\mathcal{O}}$ and $-\tilde{q}_1 = \tilde{q}_2 \equiv \tilde{q}$, we find

$$
\langle \mathcal{O}(w_1) O(w_2) \rangle = \frac{e^{-(\eta_{\mathcal{O}}\tilde{q} + \frac{k}{4}\eta_{\mathcal{O}}^2)w_{12}}}{(1 - e^{-w_{12}})^{\frac{k}{2}\eta_{\mathcal{O}}^2 + 2\eta_{\mathcal{O}}\tilde{q}}} \cdot \left( e^{\frac{w_{12}}{2}} - e^{-\frac{w_{12}}{2}} \right)^{-2\tilde{h}} = \left( e^{\frac{w_{12}}{2}} - e^{-\frac{w_{12}}{2}} \right)^{-2(\tilde{h} + \eta_{\mathcal{O}}\tilde{q} + \frac{k}{4}\eta_{\mathcal{O}}^2)} ,
\tag{2.56}
$$

which is the correct result, including all the normalizations. A similar computation hols on the right.

---

[13]This observation will be particularly relevant in $J\bar{T}$ - deformed CFTs, where the spectral flow operator does not commute with the modes of the current, and therefore it is important to establish whether it is the operator or its eigenvalue that appears in the definition of $\mathcal{O}$.

[14]Note that the action of a charge operator, e.g. $J_0$, on an out state yields *minus* its charge, since $J_0|h, q\rangle = q|h, q\rangle \Rightarrow \langle h, -q|J_0 = q\langle h, -q| = -(-q)\langle h, -q|$.

We can use the same kind of manipulations to compute the three-point function

$$\langle \mathcal{O}_1(w_1)\mathcal{O}_2(w_2)\mathcal{O}_3(w_3)\rangle = e^{-\sum_{i=1}^3 (\frac{k}{4}\eta_i^2 + \eta_i^a \tilde{q}_i^a)w_i} \langle \tilde{\mathcal{O}}_1(w_1) e^{-\eta_1^a \sum \frac{1}{n} e^{-nw_1}\tilde{J}_n^a} e^{(\eta_a q_2^a + \eta_2^a \tilde{J}_0^a)w_2 + \eta_2^a \sum \frac{1}{n} e^{nw_2}\tilde{J}_{-n}^a}$$
$$\times \tilde{\mathcal{O}}_2(w_2) e^{-\eta_2^a \sum \frac{1}{n} e^{-nw_2}\tilde{J}_n^a} e^{(\eta_a q_3^a + \eta_3^a \tilde{J}_0^a)w_3 + \eta_3^a \sum \frac{1}{n} e^{nw_3}\tilde{J}_{-n}^a} \tilde{\mathcal{O}}_3(w_3)\rangle. \tag{2.57}$$

Inside this correlator, the operators $\eta_a, \tilde{J}_0^a$ have the following eigenvalues, from left to right: $\eta^a = \eta_2^a + \eta_3^a = -\eta_1^a$, $\tilde{J}_0^a = -\tilde{q}_1^a$ and $\eta^a = \eta_3^a$, $\tilde{J}_0^a = \tilde{q}_3^a$. Upon commuting the current modes through, we find

$$\langle \mathcal{O}_1(w_1)\mathcal{O}_2(w_2)\mathcal{O}_3(w_3)\rangle =$$
$$e^{-\frac{k\eta_1^2}{4}w_1 - \frac{k\eta_2^2}{4}w_2 + \frac{k\eta_3^2}{4}w_3 - \eta_1^a\tilde{q}_1^a w_1 - \eta_1^a q_2^a w_2 + \eta_2^a\tilde{q}_3^a w_2 + \eta_3^a\tilde{q}_3^a w_3}(1 - e^{-w_{12}})^{(\eta_2^a\tilde{q}_1^a + \eta_1^a\tilde{q}_2^a + \frac{k}{2}\eta_1^a\eta_2^a)}$$
$$\times (1 - e^{-w_{23}})^{\eta_3^a\tilde{q}_2^a + \eta_2^a\tilde{q}_3^a + \frac{k}{2}\eta_2^a\eta_3^a}(1 - e^{-w_{13}})^{\eta_3^a\tilde{q}_1^a + \eta_1^a\tilde{q}_3^a + \frac{k}{2}\eta_1^a\eta_3^a}\langle \tilde{\mathcal{O}}_1(w_1)\tilde{\mathcal{O}}_2(w_2)\tilde{\mathcal{O}}_3(w_3)\rangle. \tag{2.58}$$

The expected form of this correlator is that of a primary three-point function on the cylinder, namely

$$\langle \mathcal{O}_1(w_1)\mathcal{O}_2(w_2)\mathcal{O}_3(w_3)\rangle \sim \frac{e^{w_1 h_1 + w_2 h_2 + w_3 h_3}}{(e^{w_1} - e^{w_2})^{h_1 + h_2 - h_3}(e^{w_2} - e^{w_3})^{h_2 + h_3 - h_1}(e^{w_1} - e^{w_3})^{h_1 + h_3 - h_2}}$$
$$= \frac{e^{-w_1 h_1 + w_2(h_1 - h_3) + w_3 h_3}}{(1 - e^{-w_{12}})^{h_1 + h_2 - h_3}(1 - e^{-w_{23}})^{h_2 + h_3 - h_1}(1 - e^{-w_{13}})^{h_1 + h_3 - h_2}}, \tag{2.59}$$

where the dimensions are given by (2.19). It is easy to check that the exponents of the $(1 - e^{-w_{ij}})$ factors in (2.58) precisely match these, since

$$h_1 + h_2 - h_3 = \tilde{h}_1 + \tilde{h}_2 - \tilde{h}_3 - \eta_1^a\tilde{q}_2^a - \eta_2^a\tilde{q}_1^a - \frac{k}{2}\eta_1^a\eta_2^a \tag{2.60}$$

and cyclic permutations thereof, where we used charge conservation $\tilde{q}_3^a = -\tilde{q}_1^a - \tilde{q}_2^a$, $\eta_3^a = -\eta_1^a - \eta_2^a$. Moreover, it turns out that all the prefactors in (2.58) combine precisely into the numerator of (2.59), with the end result that the three-point function of primaries in the $J^1 \wedge J^2$ - deformed CFT is given by (2.59) times the OPE coefficient, $\tilde{C}_{123}$, in the *undeformed* CFT.

We thus find that, while the conformal dimensions and charges shift as in (2.19), (2.18), the OPE coefficients, which contain the dynamical information of the theory, are *unchanged* by the deformation

$$C_{ABC} = \tilde{C}_{ABC}. \tag{2.61}$$

Finally, we work out the four-point function, with the result

$$\langle \mathcal{O}_1(w_1)\mathcal{O}_2(w_2)\mathcal{O}_3(w_3)\mathcal{O}_4(w_4)\rangle = e^P \prod_{i<j}(1 - e^{-w_{ij}})^{\eta_j^a\tilde{q}_i^a + \eta_i^a\tilde{q}_j^a + \frac{k}{2}\eta_i^a\eta_j^a}\langle \tilde{\mathcal{O}}_1(w_1)\tilde{\mathcal{O}}_2(w_2)\tilde{\mathcal{O}}_3(w_3)\tilde{\mathcal{O}}_4(w_4)\rangle, \tag{2.62}$$

where the prefactor comes from evaluating the operators $A_i$ inside the correlator, and reads

$$P = -\sum_i (\frac{k}{4}\eta_i^2 + \eta_i^a\tilde{q}_i^a)w_i - (\eta_1^a q_2^a + \eta_2^a\tilde{q}_1^a)w_2 + [(\eta_3^a + \eta_4^a)q_3^a + \eta_3^a(\tilde{q}_3^a + \tilde{q}_4^a)]w_3 + (\eta_4^a q_4^a + \eta_4^a\tilde{q}_4^a)w_4. \tag{2.63}$$

Using this, one can check that the expression for the four-point function can be simplified to

$$\langle \mathcal{O}_1(w_1)\mathcal{O}_2(w_2)\mathcal{O}_3(w_3)\mathcal{O}_4(w_4)\rangle = \prod_{i<j}\left(e^{\frac{w_{ij}}{2}} - e^{-\frac{w_{ij}}{2}}\right)^{\frac{2}{k}(q_i q_j - \tilde{q}_i\tilde{q}_j)}\langle \tilde{\mathcal{O}}_1(w_1)\tilde{\mathcal{O}}_2(w_2)\tilde{\mathcal{O}}_3(w_3)\tilde{\mathcal{O}}_4(w_4)\rangle. \tag{2.64}$$

It is also easy to see, using charge conservation, that any correlation function of the primary operators $\mathcal{O}(w_i)$ will be related through a factor of exactly the same form to the corresponding correlator in the undeformed CFT. Thus, the correlation functions of primary operators in the deformed theory can be rather trivially expressed in terms of the undeformed correlators.

Note that the expression (2.64) for the four-point function is crossing symmetric, assuming crossing symmetry of the original CFT correlator. It is interesting to rewrite this result in the language of conformal partial waves, using the fact that in a two-dimensional CFT that possesses an affine $U(1)$ symmetry, the Virasoro-Kac-Moody conformal partial wave[15] $\mathcal{W}_{h,q}(z_i, h_i, q_i, c)$ can be written as a spectral-flow-invariant Virasoro block contribution $\mathcal{V}_{\hat{h}}$ times an affine $U(1)$ block, $\mathcal{U}$

$$\mathcal{W}_{h,q}(z_i, h_i, q_i, c) = \mathcal{U}(z_i, q_i)\,\mathcal{V}_{\hat{h}}(z_i, \hat{h}_i, c-1)\,. \tag{2.65}$$

This was shown in [43] for the case of a neutral exchanged operator, and in [41, 42] in the general case. In the above, $\hat{h}_i = h_i - q_i^2/k$ are the spectral-flow invariant pieces of the conformal dimensions and the affine $U(1)$ block is given by

$$\mathcal{U}(z_i, q_i) = \prod_{i<j} z_{ij}^{\frac{2q_i q_j}{k}}\,. \tag{2.66}$$

In a given channel, the four-point function can be written as an infinite sum over conformal partial waves corresponding to the particular Virasoro-Kac-Moody representations being exchanged

$$\langle \mathcal{O}_1(z_1)\mathcal{O}_2(z_2)\mathcal{O}_3(z_3)\mathcal{O}_4(z_4)\rangle = \sum_{h,q} C_{12h} C_{34h} \mathcal{W}_{h,q}(z_i, h_i, q_i, c)\,. \tag{2.67}$$

Since the effect of the $J^1 \wedge J^2$ deformation is to induce a charge-dependent spectral flow transformation that leaves $\hat{h}$ invariant, the only change in the Virasoro-Kac-Moody blocks will come from the change in $\mathcal{U}(z_i, q_i)$, which only depends on the charges of the external operators. Thus, the change in the conformal partial waves is

$$\mathcal{W}_{h,q}(z_i, h_i, q_i, c) \xrightarrow{J^1 \wedge J^2} \prod_{i<j} z_{ij}^{\frac{2q_i q_j}{k} - \frac{2\tilde{q}_i \tilde{q}_j}{k}} \mathcal{W}_{h,q}(z_i, h_i, q_i, c)\,, \tag{2.68}$$

irrespectively of which operator is being exchanged. Mapping this result from the plane to the cylinder and using the fact that the OPE coefficients are unchanged by the deformation, we can immediately reproduce the change (2.64) in the four-point function. This is another way to check that crossing symmetry is satisfied. The bootstrap equations of the deformed CFT are thus trivially solved, given their solution in the undeformed CFT [41].

# 3 Primary operators in $J\bar{T}$ - deformed CFTs

Armed with our understanding of the $J^1 \wedge J^2$ - deformed primaries in a language that is in principle generalizable to $J\bar{T}$, we would now like to present our proposal for defining primary operators in $J\bar{T}$ - deformed CFTs. To set up the stage, we start with a few general remarks about the similarities and differences between the $J^1 \wedge J^2$ and $J\bar{T}$ cases.

---

[15]Here, $z_i, h_i, q_i$ are the positions, dimensions and respectively $U(1)$ charges of the external operators, $h, q$ are the dimension and charge (constrained by conservation) of the primary on whose family we project, and $c$ is the central charge of the CFT.

## 3.1 Setup and general remarks

Let us summarize our current understanding of $J\bar{T}$ - deformed CFTs that is relevant for this question. On the cylinder, the $J\bar{T}$ - deformed energy eigenstates flow according to

$$\partial_\lambda |n_\lambda\rangle = \mathcal{X}_{J\bar{T}}|n\rangle_\lambda\,, \tag{3.1}$$

where $\lambda$ now represents the $J\bar{T}$ flow parameter and $\mathcal{X}_{J\bar{T}}$ is a presumably well-defined operator that was worked out in [12] at the classical level, and discussed in [11] at the quantum level. Given $\mathcal{X}_{J\bar{T}}$, one can define two commuting sets of Virasoro-Kac-Moody generators $\tilde{L}_n, \tilde{J}_n$ and $\tilde{\bar{L}}_n, \tilde{\bar{J}}_n$ via the flow equation

$$\partial_\lambda \widetilde{L}_n = [\mathcal{X}_{J\bar{T}}, \widetilde{L}_n] \tag{3.2}$$

etc., which can be shown to be conserved [11]. This flow equation implies that primary states in the undeformed CFT will flow to primaries with respect to the $\tilde{L}_n$ and that the eigenvalues of $\tilde{L}_0$, etc., are independent of $\lambda$ [35], and thus equal the corresponding eigenvalues in the undeformed CFT

$$\tilde{L}_0 |h_\lambda\rangle = \tilde{h}|h_\lambda\rangle\,, \quad \tilde{J}_0 |h_\lambda\rangle = \tilde{q}|h_\lambda\rangle\,, \quad \tilde{\bar{L}}_0 |h_\lambda\rangle = \tilde{\bar{h}}|h_\lambda\rangle\,, \quad \tilde{\bar{J}}_0 |h_\lambda\rangle = \tilde{\bar{q}}|h_\lambda\rangle\,. \tag{3.3}$$

Thus, in terms of the flowed generators, the structure of the Hilbert space looks the same as that of the undeformed CFT: primary states have the same dimensions as in the seed CFT, and descendant states can be built by acting with $\widetilde{L}_{-n}$, etc. on them.

As explained in [11], the generators that implement (pseudo)conformal and affine $U(1)$ transformations in the deformed theory are given by

$$L_n = \widetilde{L}_n + \lambda H_R \widetilde{J}_n + \frac{\lambda^2 k H_R^2}{4}\delta_{n,0}\,, \qquad J_n = \tilde{J}_n + \frac{\lambda k}{2}H_R\,\delta_{n,0}\,,$$
$$\bar{L}_n = \widetilde{\bar{L}}_n + \lambda :H_R\widetilde{\bar{J}}_n: + \frac{\lambda^2 k H_R^2}{4}\delta_{n,0}\,, \qquad \bar{J}_n = \tilde{\bar{J}}_n + \frac{\lambda k}{2}H_R\,\delta_{n,0}\,, \tag{3.4}$$

whose relation to the $\tilde{L}_n$ resembles a spectral flow whose parameter is proportional to the right-moving Hamiltonian. More specifically, the left-moving generators $L_n, J_n$ implement usual conformal and affine $U(1)$ transformations, whereas the right-moving generators $\bar{L}_n, \bar{J}_n$ (with the exception of the global generators $\bar{L}_0, \bar{J}_0$) implement field-dependent conformal (1.3) and affine $U(1)$ transformations. This structure is entirely analogous to the one we have uncovered in $J^1 \wedge J^2$ - deformed CFTs; note, however, that now the spectral flow "parameter" does not commute with the modes of the right-moving currents, which adds a layer of complication to the problem.

Given our understanding of the states and symmetry generators on the cylinder, the question is how to define a physical primary operator, $\mathcal{O}$, and construct its correlation functions. Since the theory is local and conformal on the left, $\mathcal{O}$ should obey the standard primary condition with respect to the left-moving generators $L_n, J_n$. The non-trivial part of our task is to find an appropriate notion of a "primary condition" also on the non-local right-moving side. This problem does not have a counterpart in $J^1 \wedge J^2$ - deformed CFTs, where the Ward identities that primary operators must satisfy are simply determined by the fact that the deformed theory stays a CFT.

Since $H_R$ - the generator of right-moving translations - enters the relation (3.4) between the two sets of generators, it makes sense to choose a basis of operators that diagonalizes its action, i.e. work in momentum space. This is also instructed by the non-locality of the deformed CFT,

and in particular the fact that the left primary dimensions depend on the eigenvalue of this operator as in (1.2). Thus, a first difference with the $J^1 \wedge J^2$ - case is that we are forced to work in momentum space, at least as far as the right-movers are concerned. For uniformity reasons, we will choose this basis on both sides.

As in the previous section, we will construct candidate primary operators, $\mathcal{O}$, based on an auxiliary operator $\tilde{\mathcal{O}}$ that is defined to flow with the $J\bar{T}$ parameter in the same way as the energy eigenstates. As before, we will attempt to relate the vacuum correlation functions of the candidate $\mathcal{O}$ to those of $\tilde{\mathcal{O}}$, which are identical to the correlation functions in the undeformed CFT, as implied by the flow equation. Note, however, that now the 'flowed vacuum' state $|0_\lambda\rangle = U_\lambda |0\rangle_{\lambda=0}$, to which the above argument applies, is not annihilated by one of the global $SL(2,\mathbb{R})_L$ generators on the cylinder[16]. More specifically, the flow equation implies that $\tilde{L}_{-1}|0_\lambda\rangle = 0$ which, in terms of the unflowed generators, translates into

$$L_{-1}|0_\lambda\rangle = \lambda J_{-1} H_R |0_\lambda\rangle = \frac{2}{\lambda k}\left(R - \sqrt{R^2 + \frac{\lambda^2 kc}{24}}\right) J_{-1}|0_\lambda\rangle, \tag{3.5}$$

where we used the known result [27] for the flowed finite-size energy eigenvalues (in this case, the ground state). Thus, on the cylinder, the flowed vacuum is not $SL(2,\mathbb{R})$ invariant. While we see no obvious reason that an $SL(2,\mathbb{R})$ - invariant vacuum should not exist, this state will clearly be different from $|0_\lambda\rangle$.

The reason we would like the vacuum to be annihilated by the $SL(2,\mathbb{R})_L$ generators is that only then do we expect the primary correlation functions to take the standard form dictated by conformal symmetry. It is therefore important to construct this state of *a priori* greater physical interest in $J\bar{T}$ - deformed CFTs on the cylinder[17]. Rather than addressing this interesting problem - which appears somewhat complicated - we will simply avoid it by taking the $R \to \infty$ limit, which forces the two candidate vacuum states to coincide.

Our approach can thus be summarized as follows: given that out best understanding of the states, symmetry generators and their flow is on the cylinder, we will present our general construction of the candiate primary operators and their correlation functions in this setting. However, this construction is only expected to yield results consistent with $SL(2,\mathbb{R})$ invariance (and its right-moving analogue) in the $R \to \infty$ limit, where one is effectively working on the plane. This limit will turn out to also resolve a consistency problem that we will encounter along the way.

As we already mentioned, due to the non-locality of the model, we need to work in momentum space. We will thus start by reviewing some basic results about momentum-space Ward identities for primary fields in a CFT, before presenting our proposal for the momentum-space primary operators.

## 3.2 CFT Ward identities in momentum space

Let us start by introducing the momentum-space operators

$$\mathcal{O}(p) = \int dw \, e^{-pw} \mathcal{O}(w), \tag{3.6}$$

---

[16]This problem does not appear in $J^1 \wedge J^2$ - deformed CFTs, because the $\eta^a$ in (2.34) does annihilate the flowed vacuum.

[17]By definition, this state would be annihilated by $L_{-1}$, which would translate, as above, into an explicitly $\lambda$ - dependent relation between the action of $\tilde{L}_{-1}$ and $\tilde{J}_{-1}$ on it. This in turn implies that this state will not satisfy a simple flow equation involving $\mathcal{X}_{J\bar{T}}$, though one may be able to explicitly write it in the basis (3.1), using $\lambda$ - dependent coefficients. This state would also need to satisfy an appropriate constraint with respect to the global right-moving generators, whose algebra - which is not $SL(2,\mathbb{R})$ - is listed in appendix C.

where $w = i(t + \sigma)$ corresponds to a Lorentzian coordinate on the cylinder. There is, as always, also a right-moving contribution, with mometum $\bar{p}$, so that the spatial momentum $p - \bar{p}$ is quantized. Since all formulae in this subsection are identical for the right-movers, we will omit writing them explicitly.

Momentum-space CFT correlation functions can be computed by either taking the Fourier transform of the corresponding position-space Wightman functions [44, 46], or by solving the conformal Ward identities directly in momentum space [45]. While these studies were concerned with momentum-space correlators of the CFT on the plane, for the problem at hand we are interested in the form of the momentum-space conformal Ward identities on the cylinder. These can be obtained by Fourier-transforming the position-space commutators (2.49) of the $L_n, J_n$, with the result

$$[L_n, \mathcal{O}(p)] = (n(h-1) + pR)\mathcal{O}\left(p - \frac{n}{R}\right), \quad [J_n, \mathcal{O}(p)] = q\,\mathcal{O}\left(p - \frac{n}{R}\right). \tag{3.7}$$

Despite their unusual form, these can in principle be used to fix the form of the correlation functions[18].

While it would certainly be interesting to further explore the properties of the solutions to these Ward identities on the cylinder, here we are mostly interested in the limit $R \to \infty$, where they should reduce to the well-known momentum-space Ward identities on the plane. To show how this comes about, we need to relate the (dimensionless) $SL(2, \mathbb{R})$ generators on the cylinder, $L_n = -R\,e^{\frac{nw}{R}}\partial_w$ for $n = \pm 1, 0$ to the planar ones, $L_n^{pl} = -w^{n+1}\partial_w$, as $R \to \infty$. This can be simply achieved by expanding the cylinder generators at large $R$

$$L_0 = -R\,\partial_w = RL_{-1}^{pl}, \quad L_{\pm 1} = -R\,\partial_w \mp w\partial_w - \frac{w^2}{2R}\partial_w + \mathcal{O}(1/R^2). \tag{3.10}$$

The inverse relation reads

$$L_{-1}^{pl} = \frac{1}{R}L_0, \quad L_0^{pl} = \frac{1}{2}(L_1 - L_{-1}) + \mathcal{O}(1/R^2), \quad L_1^{pl} = R(L_1 + L_{-1} - 2L_0) + \mathcal{O}(1/R). \tag{3.11}$$

Expanding the momentum-space commutators (3.7) of the operator $\mathcal{O}(p)$ with the $L_n$, we find

$$[L_{-1}^{pl}, \mathcal{O}(p)] = p\,\mathcal{O}, \quad [L_0^{pl}, \mathcal{O}(p)] = (h-1)\mathcal{O} - p\,\partial_p\mathcal{O} + \mathcal{O}(1/R^2),$$
$$[L_1^{pl}, \mathcal{O}(p)] = p\,\partial_p^2\mathcal{O} + 2(1-h)\partial_p\mathcal{O} + \mathcal{O}(1/R^2), \tag{3.12}$$

exactly as expected. One can check that the solution to these Ward identities, e.g. for the two-point function, agrees with the $R \to \infty$ limit of its cylinder counterpart (3.9).

## 3.3 A proposal for primary operators in $J\bar{T}$ - deformed CFTs

We are now ready to construct a set of momentum-space operators in $J\bar{T}$ - deformed CFTs that come as close as possible to being primary, in the sense of (3.7). As we explained, we will start by working on the cylinder and then take the $R \to \infty$ limit in which, at least for a CFT, the

---

[18]For example, for a two-point function, they give a constraint of the form

$$0 = \langle 0|L_1 L_{-1}\mathcal{O}_1(p_1)\mathcal{O}_2(p_2)|0\rangle = [(1 - h_1 + p_1 R)(h_1 + p_1 R) + (1 - h_2 + p_2 R)(h_2 + p_2 R)]\mathcal{O}_1(p_1)\mathcal{O}_2(p_2) \tag{3.8}$$
$$+ (1 - h_1 + p_1 R)(h_2 - 1 + p_2 R)\mathcal{O}(p_1 + 1/R)\mathcal{O}(p_2 - 1/R) + (1 - h_2 + p_2 R)(h_1 + p_1 R - 1)\mathcal{O}(p_1 - 1/R)\mathcal{O}(p_2 + 1/R),$$

where two generator insertions are necessary in order to respect momentum conservation. A solution to the above constraint is

$$\langle \mathcal{O}_1(p_1)\mathcal{O}_2(p_2)\rangle \sim e^{\pm i\pi R(p_1 - p_2)/2}\Gamma(h_1 + p_1 R)\Gamma(h_2 + p_2 R), \quad \text{where} \quad p_1 + p_2 = 0, \tag{3.9}$$

which is proportional to the Fourier transform of the position space two-point function.

momentum-space Ward identities and their solutions seamlessly translate from the cylinder to the plane.

We start by introducing again a set of operators $\tilde{\mathcal{O}}(w, \bar{w})$ that formally satisfy the flow equation

$$\partial_\lambda \tilde{\mathcal{O}}(w, \bar{w}) = [\mathcal{X}_{J\bar{T}}, \tilde{\mathcal{O}}(w, \bar{w})], \tag{3.13}$$

with the initial condition that they equal the local operator $\mathcal{O}(w, \bar{w})$ in the undeformed CFT. This definition automatically implies that the correlation functions of $\tilde{\mathcal{O}}(w, \bar{w})$ will be identical to those in the undeformed CFT, irrespectively of the non-locality of the deformed theory, provided we evaluate them in the flowed vacuum state, $|0_\lambda\rangle$. We re-emphasize that these operators - previously discussed in [33] - are just auxiliary, formal constructs with no particular physical significance, as they do not correspond to local operators even on the local, left-moving side. In particular, $w, \bar{w}$ are simply labels, corresponding to the position of the initial local CFT operator that we flow, which have no particular meaning in the deformed theory.

Since both $\tilde{\mathcal{O}}(w, \bar{w})$ and $\tilde{L}_n$, etc. flow in the same way with $\mathcal{X}_{J\bar{T}}$, it follows that $\tilde{\mathcal{O}}(w, \bar{w})$ will satisfy the usual Ward identities

$$[\tilde{L}_n, \tilde{\mathcal{O}}(w, \bar{w})] = e^{nw}[n\tilde{h}\tilde{\mathcal{O}}(w, \bar{w}) + \partial_w \tilde{\mathcal{O}}(w, \bar{w})], \quad [\tilde{J}_n, \tilde{\mathcal{O}}(w, \bar{w})] = e^{nw}\tilde{q}\,\tilde{\mathcal{O}}(w, \bar{w}),$$

$$[\bar{\tilde{L}}_n, \tilde{\mathcal{O}}(w, \bar{w})] = e^{n\bar{w}}[n\bar{\tilde{h}}\tilde{\mathcal{O}}(w, \bar{w}) + \partial_{\bar{w}} \tilde{\mathcal{O}}(w, \bar{w})], \quad [\bar{\tilde{J}}_n, \tilde{\mathcal{O}}(w, \bar{w})] = e^{n\bar{w}}\bar{\tilde{q}}\,\tilde{\mathcal{O}}(w, \bar{w}), \tag{3.14}$$

where, again, $w, \bar{w}$ are just labels inherited from the undeformed theory.

As explained at the beginning of this section, our candidate primary operators should be constructed in momentum space, and so they satisfy

$$[H_R, \mathcal{O}(p, \bar{p})] = \bar{p}\,\mathcal{O}(p, \bar{p}). \tag{3.15}$$

In addition, they should be primary with respect to the unflowed left-moving generators $L_n, J_n$

$$L_n = \tilde{L}_n + \eta \tilde{J}_n + \frac{k\eta^2}{4}\delta_{n,0}, \quad J_n = \tilde{J}_n + \frac{k\eta}{2}\delta_{n,0}, \quad \eta \equiv \lambda H_R, \tag{3.16}$$

with the expected eigenvalues

$$h = \tilde{h} + \eta_\mathcal{O}\tilde{q} + \frac{k\eta_\mathcal{O}^2}{4}, \quad q = \tilde{q} + \frac{k}{2}\eta_\mathcal{O}, \quad \eta_\mathcal{O} = \lambda\bar{p}. \tag{3.17}$$

This simply translates into the constraint (3.7), where $h, q$ are given above. The reason we introduced the above notation is to highlight the similitude with the $J^1 \wedge J^2$ example.

Given our experience with the $J^1 \wedge J^2$ deformation, we can easily construct a solution for the left-moving piece of $\mathcal{O}(p, \bar{p})$ that satisfies (3.7). As for its right-moving piece, we expect a similar constraint to hold in terms of the right-moving generators $\bar{L}_n, \bar{J}_n$; however, given that these are not standard conformal generators, we do not know exactly what relation to impose. We thus resort to simply guessing an appropriate right-moving factor, work out its properties, and then justify our choice *a posteriori* via its rather reasonable predictions in the $R \to \infty$ limit.

Our proposed definition of an operator that has the required commutation relations with the left-moving generators and possibly reasonable commutators with the right-moving ones is

$$\mathcal{O}(p, \bar{p}) = \int dw d\bar{w}\, e^{-pw - \bar{p}\bar{w}} \epsilon^{A_\mathcal{O}w + B_\mathcal{O}\bar{w}} e^{\eta_\mathcal{O} \sum_{n=1}^{\infty} \frac{1}{n}(e^{nw}\tilde{J}_{-n} + e^{n\bar{w}}\bar{\tilde{J}}_{-n})} \tilde{\mathcal{O}}(w, \bar{w}) e^{-\eta_\mathcal{O} \sum_{n=1}^{\infty} \frac{1}{n}(e^{-nw}\tilde{J}_n + e^{-n\bar{w}}\bar{\tilde{J}}_n)}, \tag{3.18}$$

where $A_{\mathcal{O}}$ and $B_{\mathcal{O}}$ are operators - which we will determine shortly - that depend on the conserved quantum numbers of $\mathcal{O}$ and - as in the $J^1 \wedge J^2$ case - are assumed to be linear combinations of $\tilde{J}_0, \tilde{\bar{J}}_0$ and $H_R$. Note that, unlike in the $J^1 \wedge J^2$ example, these operators do not commute with exponential factor that follows. The split between $A_{\mathcal{O}}, B_{\mathcal{O}}$ and the $p, \bar{p}$ factors we pulled out is of course completely arbitrary, but will be soon fixed in a convenient fashion. For the time being, we are still working on the cylinder and have set $R = 1$; the factor of the radius can be easily reinstated by dimensional analysis: $w \to w/R$ and $p \to pR$, with $\eta$ fixed.

Given this explicit expression, we can simply compute the commutators of $\mathcal{O}(p, \bar{p})$ with the various generators. The simplest such commutator is with the left-moving current, which reads

$$[J_n, \mathcal{O}(p, \bar{p})] = \left(\tilde{q} + \frac{k}{2}\eta_{\mathcal{O}}\right)\mathcal{O}(p - n, \bar{p}) \tag{3.19}$$

exactly as expected. Note that for $n \neq 0$, the shift in the charge comes from the coefficient of $\tilde{J}_n$ in the exponent of (3.18), which therefore needs to be a number. For $n = 0$, it comes from the commutator with $H_R$, which thus sets $\eta_{\mathcal{O}} = \lambda\bar{p}$, with $\bar{p}$ defined in (3.15).

The commutator with $L_n$ reads

$$\begin{aligned}[L_n, \mathcal{O}(p, \bar{p})] = &\left[n\left(\tilde{h} + \eta_{\mathcal{O}}\tilde{q} + \frac{k\eta_{\mathcal{O}}^2}{4} - 1\right) + p\right]\mathcal{O}(p - n, \bar{p}) \\ &+ \left(\eta_{\mathcal{O}}\tilde{J}_0 + \eta q - A_{\mathcal{O}} - \eta_{\mathcal{O}}\tilde{q} - \frac{k\eta_{\mathcal{O}}^2}{4}\right)\mathcal{O}(p - n, \bar{p}) \\ &+ (\lambda\bar{p} - \eta_{\mathcal{O}})\mathcal{O}(p, \bar{p})\tilde{J}_n, \end{aligned} \tag{3.20}$$

where the last term vanishes, for the reason we just stated. The primary condition with respect to $L_n$ then fixes

$$A_{\mathcal{O}} = \eta_{\mathcal{O}}\tilde{J}_0 + \eta q - \eta_{\mathcal{O}}\tilde{q} - \frac{k}{4}\eta_{\mathcal{O}}^2, \tag{3.21}$$

which is entirely analogous to the expression (2.52) we obtained in $J^1 \wedge J^2$ - deformed CFTs.

We now turn to the commutation relations with the right-movers, still for $R$ finite. The commutation relations with the right-moving $U(1)$ current are

$$[\bar{J}_n, \mathcal{O}(p, \bar{p})] = \left(\tilde{\bar{q}} + \frac{k\eta_{\mathcal{O}}}{2}\right)\mathcal{O}(p, \bar{p} - n) + \tilde{\bar{J}}_n[\mathcal{O}(p, \bar{p}) - \mathcal{O}(p + a\alpha_n^r, \bar{p} + b\alpha_n^r)], \tag{3.22}$$

where $a, b$ are the coefficients of $H_R$ inside $A_{\mathcal{O}}, B_{\mathcal{O}}$ and $\alpha_n$, given in (C.2), is defined through the commutator

$$[\tilde{\bar{J}}_n, H_R] = \tilde{\bar{J}}_n\alpha_n. \tag{3.23}$$

The somewhat suspicious-looking operator-valued shift of the arguments in the last term is simply a shorthand for the corresponding Fourier-space expression, and follows from the contribution of terms of the form

$$\tilde{\bar{J}}_n - e^{cH_R}\tilde{\bar{J}}_n e^{-cH_R} = \tilde{\bar{J}}_n(1 - e^{-c\alpha_n}), \tag{3.24}$$

with $c = aw$ or $b\bar{w}$ to the commutator. Interestingly, this shift affects both the left and the right-moving side. Note also that (3.21) implies that $a = \lambda q$.

Notwithstanding the second term - which will turn out to be negligible at large $R$ - (3.22) takes precisely the form of a CFT Ward identity between an affine Kac-Moody current and an operator of right-moving charge

$$\bar{q} = \tilde{\bar{q}} + \frac{k\eta_{\mathcal{O}}}{2}, \tag{3.25}$$

which exactly mirrors the behaviour on the left-moving side. The fact that all the $\bar{J}_n$ modes have this behaviour, and not just the $n = 0$ one (which corresponds to the global right-moving charge), is an interesting output of our construction.

Let us now also fix the operator $B_{\mathcal{O}}$, e.g. by computing the commutator of $\mathcal{O}(p,\bar{p})$ with $\tilde{\bar{L}}_0 = H_R(R - \lambda \tilde{J}_0 - \frac{\lambda^2 k}{4} H_R)$ which, using (3.15), yields[19]

$$[\tilde{\bar{L}}_0, \mathcal{O}(p,\bar{p})] = \left(\bar{p} + \lambda \tilde{q} \bar{p} + \frac{\lambda^2 k}{4} \bar{p}^2 - \lambda \tilde{J}_0 \bar{p} - \lambda q H_R\right) \mathcal{O}(p,\bar{p}). \tag{3.26}$$

This commutator can also be evaluated by using (3.18) and the Ward identity (3.14) for $\tilde{\mathcal{O}}$, with the result

$$[\tilde{\bar{L}}_0, \mathcal{O}(p,\bar{p})] = (\bar{p} - B_{\mathcal{O}}) \mathcal{O}(p,\bar{p}). \tag{3.27}$$

Equating the two expressions, we find that the solution for $B_{\mathcal{O}}$ is identical[20] to the one for $A_{\mathcal{O}}$, (3.21). This may seem a bit surprising, as we would have expected the right-moving $B_{\mathcal{O}}$ to depend on the right-moving charges and current operators, as it does in $J^1 \wedge J^2$ - deformed CFTs (2.52), and not on the left-moving ones. The reason for this dependence can be traced back to the fact that $\bar{L}_0$ - the dimensionless right-moving generator that is related to $\tilde{\bar{L}}_0$ by spectral flow - does not equal $RH_R$, but rather $\bar{L}_0 = R_v H_R$, where $R_v = R - \lambda(J_0 - \bar{J}_0)$ is the field-dependent radius of the field-depedent right-moving coordinate. This field-dependent rescaling has no counterpart in $J^1 \wedge J^2$ - deformed CFTs, and we could not see any simple modification of (3.18) that would yield a $B_{\mathcal{O}}$ of the expected form, without spoiling the rather pleasing commutation relations we have obtained so far. One could, of course, consider working in terms of the field-dependent coordinate on the right-moving side - which natually involves factors of $R_v$ - but the resulting expressions are significantly more complicated than (3.18).

The reason that this particular expression for $B_{\mathcal{O}}$ is problematic is due to its effect on the correlation functions of $\mathcal{O}(p,\bar{p})$ on the cylinder, which will be discussed in the next subsection. However, as we will show, this effect is subleading at large $R$. Since, as we explained earlier, the $R \to \infty$ limit is also needed to resolve the problem with the choice of vacuum in $J\bar{T}$ - deformed CFTs, we will simply continue to use the proposed expression (3.18), despite its drawbacks at finite $R$, and show that it does indeed yield very reasonable predictions as $R \to \infty$.

Let us now finally compute the commutation relations of $\mathcal{O}(p,\bar{p})$ with the unflowed right-moving generators, $\bar{L}_n$, given in (3.4). We find

$$
\begin{aligned}
[\bar{L}_n, \mathcal{O}(p,\bar{p})] = & \left[n\left(\tilde{\bar{h}} + \eta_{\mathcal{O}}\tilde{\bar{q}} + \frac{k\eta_{\mathcal{O}}^2}{4} - 1\right) + \bar{p}\right]\mathcal{O}(p,\bar{p}-n) \\
& + (\tilde{\bar{L}}_n + : \eta \tilde{\bar{J}}_n :)[\mathcal{O}(p,\bar{p}) - \mathcal{O}(p + a\alpha_n, \bar{p} + b\alpha_n)] \\
& + [(\eta_{\mathcal{O}} - \eta)(\tilde{q} - \tilde{\bar{q}}) - \eta_{\mathcal{O}}(\tilde{J}_0 - \tilde{\bar{J}}_0)]\mathcal{O}(p,\bar{p}-n).
\end{aligned}
\tag{3.28}
$$

The first term on the right-hand side looks exactly like a momentum-space conformal Ward identity in a CFT, where the right-moving conformal dimension of the operator is given by

$$\bar{h} = \tilde{\bar{h}} + \eta_{\mathcal{O}}\tilde{\bar{q}} + \frac{k\eta_{\mathcal{O}}^2}{4}. \tag{3.29}$$

Using $\eta_{\mathcal{O}} = \lambda\bar{p}$, this exactly corresponds to a momentum-dependent spectral flow of the right-moving dimensions. Note that, unlike for the left-movers, this expression does *not* follow from

---

[19]Note we could have chosen the eigenvalue of $H_R$ in (3.15) to be different from the $\bar{p}$ factor appearing in the definition (3.18). This would have simply resulted in an expression for $B_{\mathcal{O}}$ that depended on both constants, as only $\bar{p} - B_{\mathcal{O}}$ is fixed.

[20]One can also check that with this choice, $\mathcal{O}(p,\bar{p})$ satisfies the expected Hermiticity conditions.

the flow of the right-moving energy eigenvalues on the cylinder, as the latter involve a factor of $\tilde{q}$, rather than $\tilde{\bar{q}}$.

The second term can be written as $\bar{L}_n$ multiplying the term in parantheses (with $a = b = \lambda q$), since the latter vanishes for $n = 0$. As we will argue, this term will drop out in the $R \to \infty$ limit. The last term is required by the definition of the right-moving generators, which include a factor of the field-dependent radius - e.g. for $n = 0$, $\bar{L}_0 = H_R R_\nu$. Since we have chosen to diagonalize $H_R$, and not $\bar{L}_0$, this term is a simple consequence of the non-trivial commutator of $\mathcal{O}$ with the winding operator appearing in $R_\nu$.

Let us now discuss the $R \to \infty$ limit of these commutators. We note that the "operator-valued shift" of $p, \bar{p}$ in (3.22) and (3.28) scales with $R$ as

$$p + \frac{\lambda q \alpha_n}{R} \approx p + \frac{\lambda q n \hbar}{R^2}, \quad \text{as} \quad R \to \infty. \tag{3.30}$$

Thus, the contribution of this term to the commutators of $\mathcal{O}$ with $\bar{L}_n, \bar{J}_n$ scale as $1/R^2$ in the large $R$ limit. It is easy to see from (3.12) that these terms will consequently not contribute to the Ward identities on the plane, at least as far as $\bar{L}_{\pm 1, 0}$ are concerned. As for the last term in (3.28), we note that it is suppressed by $1/R$ in the commutator with $\bar{L}_0/R \approx L^{pl}_{-1}$, and it drops out from the combinations $\bar{L}_1 - \bar{L}_{-1}$ and $R(\bar{L}_1 + \bar{L}_{-1} - 2\bar{L}_0)$, which are in principle identified with the plane generators.

To summarize, while at finite $R$ our candidate primary operators obey precisely CFT Ward identities with respect to the left-moving generators and certain 'CFT-like' Ward identities with respect to the right-moving ones, in the $R \to \infty$ limit all Ward identities appear to reduce to exactly CFT ones, at least as far as the global conformal and Kac-Moody generators are concerned. This is consistent with the fact that the right-moving algebra becomes Virasoro-Kac-Moody in the strict $R \to \infty$ limit. One should be cautious, however, about the presence of subtle contributions to the Ward identities in this limit - related to the momentum dependence of the conformal dimensions - and thus a more careful study is called for.

While the CFT-like form of the Ward identities (3.22) and (3.28) looks rather appealing, especially as $R \to \infty$, note that we have by no means derived it from first principles. For that, one would need to better understand how operators transform under the pseudo-conformal symmetries generated by $\bar{L}_n, \bar{J}_n$, taking into account the fact that their algebra is not Virasoro-Kac-Moody at finite $R$. One can alternatively use the Ward identities we worked out as a *definition* of what is to be meant by a primary operator in the non-local $J\bar{T}$ - deformed CFT, whose solution is, of course, (3.18). However, it seems hard to justify the choice (3.22), (3.28), especially at finite $R$. It is, nevertheless, intriguing that our candidate primary operators (3.18) satisfy such simple-looking Ward identities with respect to the pseudo-conformal generators, which likely hint towards a much richer, pseudo-local structure of $J\bar{T}$ - deformed CFTs. This intuition is further supported by our results for the correlation functions, which are presented in the next subsection.

## 3.4 Correlation functions

The computation of correlation functions of the candidate primary operators (3.18) in $J\bar{T}$ - deformed CFTs proceeds in direct analogy to its $J^1 \wedge J^2$ counterpart, which we detailed in section 2.4. We will first evaluate the correlation functions at finite $R$, using the flowed vacuum $|0_\lambda\rangle$ - in which the correlators of the auxiliary $\tilde{\mathcal{O}}$ operators reduce to the ones in the undeformed CFT - and only take the $R \to \infty$ limit at the end. This way of proceeding will make it clear that the contribution associated with the non-CFT-like term in (3.28) drops out from the correlation function in the decompactification limit.

The momentum-space two-point function of the candidate primary operators reads

$$\langle \mathcal{O}_{-q}(p_1,\bar{p}_1)\mathcal{O}_q(p_2,\bar{p}_2)\rangle = \int d^2w_1 d^2w_2 e^{-\sum_i(p_i w_i + \bar{p}_i \bar{w}_i)} e^{-\eta_{vac}q(w_{12}+\bar{w}_{12})-\eta_{\mathcal{O}}(\tilde{q}-\bar{\tilde{q}})\bar{w}_{12}}$$

$$\times \langle \mathcal{O}_{h_1,\bar{h}_1}(w_1,\bar{w}_1)\mathcal{O}_{h_2,\bar{h}_2}(w_2,\bar{w}_2)\rangle, \tag{3.31}$$

where $\bar{p}_1 = -\bar{p}_2 = -\bar{p}$ by momentum conservation, imposed as usual by the integral over the center of mass position. The correlation function on the last line is a usual position-space CFT two-point function, with the same normalization as in the undeformed CFT, but with conformal dimensions given by

$$h_i(\bar{p}_i) = \tilde{h}_i + \lambda \tilde{q}_i \bar{p}_i + \frac{\lambda^2 k}{4}\bar{p}_i^2, \quad \bar{h}_i(\bar{p}_i) = \bar{\tilde{h}}_i + \lambda \bar{\tilde{q}}_i \bar{p}_i + \frac{k\lambda^2}{4}\bar{p}_i^2, \tag{3.32}$$

where $\tilde{h}_i, \bar{\tilde{h}}_i$ are the conformal dimensions of the respective operator in the undeformed CFT. The momentum-dependent shift in the dimensions, which corresponds to a spectral flow with parameters $\eta_i = \bar{\eta}_i = \lambda \bar{p}_i$ for each of the operators, occurs through exactly the same mechanism as for the $J^1 \wedge J^2$ two-point function (2.56). The only differences with this previous case are that: the spectral flow operator no longer annihilates the flowed vacuum, so we defined

$$\eta|0_\lambda\rangle = \lambda H_R |0_\lambda\rangle = \eta_{vac}|0_\lambda\rangle \neq 0, \tag{3.33}$$

where the actual eigenvalue can be read off from (3.5). This term contributes to the correlator as indicated above. The second difference is due to the expression (3.21) for $B_{\mathcal{O}}$ which, as discussed, depends on the left-moving $U(1)$ charges, instead of the right-moving ones. This leads to an explicit additional dependence on the winding charge, also indicated in (3.31).

We now write the $\langle \mathcal{O}_1(w_1,\bar{w}_1)\mathcal{O}_2(w_2,\bar{w}_2)\rangle$ correlator in terms of its Fourier transform, $\mathcal{G}_{h_i,\bar{h}_i}(p,\bar{p}) \times \delta^2(p_1 + p_2)$, and perform the trivial integral over $w_i,\bar{w}_i$. We obtain

$$\langle \mathcal{O}_{-q}(-p,-\bar{p})\mathcal{O}_q(p,\bar{p})\rangle = \int dp'd\bar{p}'\delta(p'-p+\eta_{vac}q)\delta(\bar{p}'-\bar{p}+\eta_{vac}q+\eta_{\mathcal{O}}(\tilde{q}-\bar{\tilde{q}}))\mathcal{G}_{h_i(\bar{p}),\bar{h}_i(\bar{p})}(p',\bar{p}')$$

$$= \mathcal{G}_{h_i(\bar{p}),\bar{h}_i(\bar{p})}\big(p-\eta_{vac}q,\bar{p}-\eta_{vac}q-\eta_{\mathcal{O}}(\tilde{q}-\bar{\tilde{q}})\big), \tag{3.34}$$

where $q$ itself depends on $\bar{p}$ as in (3.17). Thus, the momentum-space two-point function of the operators (3.18) is precisely given (up to some trivial shifts in the arguments) by a momentum-space CFT two-point function, but with the conformal dimensions replaced by their momentum-dependent counterparts[21] (3.32). Remarkably, this is exactly the same behaviour that we observed in (1.1) for scattering amplitudes off near-extremal black holes!

Let us now study the three-point function of $\mathcal{O}(p,\bar{p})$. We similarly obtain

$$\langle \mathcal{O}_1(p_1,\bar{p}_1)\mathcal{O}_2(p_2,\bar{p}_2)\mathcal{O}_3(p_3,\bar{p}_3)\rangle = \int d^2w_i \, e^{-\sum_i(w_i p_i + \bar{w}_i \bar{p}_i)} e^{\eta_{vac}\sum_i q_i(w_i+\bar{w}_i)} \tag{3.35}$$

$$\times e^{-\eta_1(q_1-\bar{q}_1)\bar{w}_1 + [\eta_2(q_3-\bar{q}_3)-\eta_1(q_2-\bar{q}_2)]\bar{w}_2 + \eta_3(q_3-\bar{q}_3)\bar{w}_3}\langle \mathcal{O}_{h_1,\bar{h}_1}(w_1,\bar{w}_1)\mathcal{O}_{h_2,\bar{h}_2}(w_2,\bar{w}_2)\mathcal{O}_{h_3,\bar{h}_3}(w_3,\bar{w}_3)\rangle,$$

where the three-point function appearing on the last line corresponds precisely to a CFT three-point function in position space, with conformal dimensions given by the momentum-dependent expressions (3.32) and is derived through exactly the same steps as for the $J^1 \wedge J^2$

---

[21]Note that when writing this expression, one should replace the dimensions by their momentum-dependent counterparts not only in the functional part of the correlator, but also in the prefactors, which contain factors of e.g. $\Gamma(h)$. This resonates with the behaviour of the correlation functions discussed in the single-trace version of $T\bar{T}$-deformed CFTs, which are computed using worldsheet string theory [47].

- deformed three-point function (2.58). Fourier-transforming this (CFT) expression and performing the $w_i, \bar{w}_i$ integrals, one again obtains a result that corresponds to the original momentum-space CFT three-point function with the operator dimensions replaced by (3.32), and with slightly shifted arguments, as in (3.34). Note this implies that the OPE coefficients, appropriately defined, do not change with the deformation, as was the case in $J^1 \wedge J^2$.

The computation of higher-point functions on the cylinder proceeds in an identical manner. In the case of the four-point function, we note that while the position-space four-point function that appears in the integrand is entirely crossing symmetric, as we showed in section 2.4 for the analogous $J^1 \wedge J^2$ deformation, the winding-dependent prefactors due to the unexpected form of $B_{\mathcal{O}}$ are not, and thus spoil the crossing symmetry of the result. Consequently, our proposal is not quite correct in finite size.

This is however easy to fix by taking the $R \to \infty$ limit. Noting that the only way in which these winding terms enter the correlator is through the shift of the argument of the momentum-space correlator, as in (3.34), it is clear that they can be dropped in the $R \to \infty$ limit, since they scale as $\eta_{\mathcal{O}}(\tilde{q}-\tilde{\bar{q}})/R$ with respect to $\bar{p}$. A similar comment applies to the $\eta_{vac} q$ term, which can also be dropped. The resulting four-point functions are simply the Fourier transform of position-space correlators of the form (2.64), which are manifestly crossing symmetric and are entirely determined by the corresponding four-point function in the undeformed CFT. These correlation functions should be considered in the $R \to \infty$ limit, in which they simply become correlators on the plane. Identical comments apply to higher-point functions.

The fact that all the correlation functions of our candidate $J\bar{T}$ primary operators are entirely determined by the original CFT correlators in such a strikingly simple manner strongly suggests that $J\bar{T}$ - deformed CFTs possess a very similar structure to that of standard two-dimensional CFTs, which simply awaits for the right language to be uncovered. We hope that some of the tools proposed in this article will be helpful in making progress on this interesting issue.

# 4 Discussion

In this article, we have argued that, despite their non-locality, $J\bar{T}$ - deformed CFTs do allow for a notion of primary operators with respect to the generators of the field-dependent symmetries that act on the non-local side. We moreover showed how to compute arbitrary correlation functions of these operators *exactly* in terms of the correlators of the undeformed CFT. These correlation functions appear consistent (i.e., they obey crossing symmetry) in the decompactification limit.

As discussed in the introduction, that special non-local theories may posses a structure that is sufficiently rigid to completely fix the form of low-point correlation functions is extremely interesting, as such theories could provide a microscopic dual to generic near-extremal (and, possibly, also non-extremal [48]) black holes. It thus seems worthwhile to better understand this structure, as well as its possible generalizations, and compare it to the results of scattering in black hole backgrounds.

A basic question is to understand from "first principles" the Ward identities that primary operators should satisfy, by relating them to the expected (position-space) transformation properties of the operator under field-dependent coordinate transformations. This "first principles" derivation should also be able to determine whether there are corrections to the primary operator that involve the field-dependent coordinate, an effect that we could perhaps not see due to the large $R$ limit.

A related task is to directly work out the general constraints that these Ward identities impose on correlation functions, i.e. without appealing to the auxiliary construction involv-

ing the $\tilde{\mathcal{O}}$ operators, but rather paralleling the usual argument used for standard CFTs. This question can be asked either on the cylinder or on the plane, and each case presents its own challenges: on the cylinder, one first needs to undestand the properties of the $SL(2,\mathbb{R})$ - invariant vacuum which, as explained, is different from the flowed one, and thus the result for the correlation functions could be rather different from those discussed in the previous section; on the plane, the momentum-space Ward identities should receive contributions from the explicit momentum dependence of the conformal dimensions, which is not clear how to recover from the $R \to \infty$ limit. Nonetheless, the final result that we have obtained for the correlation functions suggest that one should obtain as many constraints as there are in usual CFTs, though the language in which they are expressed may be different.

On the more technical side, an interesting issue that we have encountered concerns the possible definitions of a vacuum state for $J\bar{T}$ - deformed CFTs on a cylinder, and its $SL(2,\mathbb{R})$ invariance properties. For the two possible choices of vacuum we discussed in section 3.1, it would be interesting to work out their definition and exact relation, both at finite radius and in the $R \to \infty$ limit. Another interesting technical question is to understand how the construction presented in this article works in the specific case of conserved currents, for example the stress tensor, and how to express these symmetry currents in terms of the associated conserved charges. This may also clarify how the field-dependent coordinate, which is essential for the definition of the charges, may fit in with the flow equations and momentum-space picture for the operators used herein.

Finally, it would be very interesting to extend these results to other special non-local theories, such as $T\bar{T}$ - deformed CFTs. An essential input for our present construction was the existence, in $J\bar{T}$ - deformed CFTs, of two different bases for the right-moving symmetries: as a set of generators that flow in the same way as the energy eigenstates or, as the generators that directly implement pseudoconformal transformations. In $T\bar{T}$ - deformed CFTs, only the first set have been shown to exist at the full quantum level [11]; as for the generators of field-dependent symmetries, they are currently understood only at a classical level and on the plane [34]. One may nevertheless hope that the needed relation between the two will eventually be found (though, most likely, it will not correspond to a spectral flow, since the existence of a $U(1)$ current is not required in this case) and can be used to define an analogous set of primary operators. It is interesting to note that the most naïve guess - based on the analogy with $J\bar{T}$ - for how an appropriately defined 'primary' two-point function will be changed - namely, via a momentum-dependent shift in the conformal dimensions - matches the behaviour found in [29].

Other interesting extensions of this work would be to the single-trace versions of the $T\bar{T}$ and $J\bar{T}$ deformation, where one should, in addition, be able to compare the proposed definition of the primary operators with the expectation from worldsheet string theory [47, 49]. One may hope that, by extending these type of symmetries and their consequences to an ever larger class of theories, one would ultimately be able to conjecture a set of axioms (e.g., for the correlation functions) that all "dipole CFTs" - or, more generally, all "non-local CFTs" - should obey, and that this definition would be general enough to capture, in a holographic sense, the near-horizon dynamics of all extremal and non-extremal black holes.

# Acknowledgements

The author would like to thank Miguel Paulos, Sylvain Ribault and especially Balt van Rees for useful conversations. She is especially grateful to Alessandro Bombini and Andrea Galliani for collaboration on a related project. This research was supported in part by the ERC starting grant 679278 Emergent-BH.

# A $SL(2,\mathbb{R})$ and Kac-Moody generator identities

In this appendix, we derive a few identities that are useful in the manipulations of section 2.3.

- $SL(2,\mathbb{R})$

Given the $SL(2,\mathbb{R})$ generators $L_{\pm 1,0}$, which satisfy the usual commutation relations, we would like to find the relations between the coefficients $(a,b,c)$ and $(\tilde{a},\tilde{b},\tilde{c})$ in

$$e^{aL_{-1}}e^{2bL_0}e^{cL_1} = e^{\tilde{a}L_1}e^{2\tilde{b}L_0}e^{\tilde{c}L_{-1}},\tag{A.1}$$

which correspond to two different ways of parametrizing the same group element. This identity can be derived by using the following representation of $L_{\pm 1,0}$

$$L_{-1} = \begin{pmatrix} 0 & 1 \\ 0 & 0 \end{pmatrix},\qquad L_0 = \frac{1}{2}\begin{pmatrix} 1 & 0 \\ 0 & -1 \end{pmatrix},\qquad L_1 = \begin{pmatrix} 0 & 0 \\ -1 & 0 \end{pmatrix}.\tag{A.2}$$

Using this, we find

$$e^{aL_{-1}}e^{2bL_0}e^{cL_1} = \begin{pmatrix} 1 & a \\ 0 & 1 \end{pmatrix}\begin{pmatrix} e^b & 0 \\ 0 & e^{-b} \end{pmatrix}\begin{pmatrix} 1 & 0 \\ -c & 1 \end{pmatrix} = \begin{pmatrix} e^b - ace^{-b} & ae^{-b} \\ -ce^{-b} & e^{-b} \end{pmatrix},\tag{A.3}$$

and

$$e^{\tilde{a}L_1}e^{2\tilde{b}L_0}e^{\tilde{c}L_{-1}} = \begin{pmatrix} e^{\tilde{b}} & \tilde{c}e^{\tilde{b}} \\ -\tilde{a}e^{\tilde{b}} & e^{-\tilde{b}} - \tilde{a}\tilde{c}e^{\tilde{b}} \end{pmatrix}.\tag{A.4}$$

This leads to the relation

$$b = -\ln(e^{-\tilde{b}} - \tilde{a}\tilde{c}e^{\tilde{b}}),\qquad a = \frac{\tilde{c}}{e^{-2\tilde{b}} - \tilde{a}\tilde{c}},\qquad c = \frac{\tilde{a}}{e^{-2\tilde{b}} - \tilde{a}\tilde{c}}.\tag{A.5}$$

We also note the identity

$$e^{\alpha L_1 + 2\beta L_0 + \gamma L_{-1}} \equiv e^M = \cosh(\sqrt{\det M})I + \sinh(\sqrt{\det M})\frac{M}{\sqrt{\det M}},\tag{A.6}$$

which we can use to show that ($z = Re\,z + iIm\,z = |z|e^{i\phi}$)

$$e^{zL_{-1} - \bar{z}L_1} = e^{e^{i\phi}\tanh|z|L_{-1}}e^{-2L_0 \ln\cosh|z|}e^{-e^{-i\phi}\tanh|z|L_1}.\tag{A.7}$$

Thus, if we let $e^w = e^{i\phi}\tanh|z|$ and act on $|h\rangle$, we obtain precisely (2.37), up to certain overall factors coming from the action of the middle term. Note that $|e^w| < 1$ or $Re\,w < 0$, which implies that this interpretation only applies to operators that are to the past of the $\tau = 0$ slice, where the state is defined.

Another possibly useful identity is

$$e^{\alpha L_0}e^{e^w L_{-1}} = e^{e^{w+\alpha}L_{-1}}e^{\alpha L_0},\tag{A.8}$$

which can be used to check that $L_0$ implements a translation of the state created by acting with $\mathcal{O}(w)$ on the vacuum $\mathcal{O}(w)|0\rangle = e^{wh}e^{e^w L_{-1}}|h\rangle$.

- $SL(2, \mathbb{R})$ **- Kac-Moody**

A more interesting identity to derive is the following

$$e^{e^w(\tilde{L}_{-1} + \eta \tilde{J}_{-1})} e^{-e^w \tilde{L}_{-1}} = e^{\eta \sum_{n=1}^{\infty} \frac{1}{n} e^{nw} \tilde{J}_{-n}} \tag{A.9}$$

claimed in the same section, where $\eta$ is a constant or an operator that commutes with all other operators that appear in this expression.

To prove this, we first use the Baker-Campbell-Hausdorff formula, which states that

$$e^X e^Y = e^{\sum_{n=1}^{\infty} \frac{1}{n} (-1)^{n-1} \sum_{r_i, s_i} c(r_i, s_i) [X^{r_1} Y^{s_1} \dots X^{r_n} Y^{s_n}]}, \tag{A.10}$$

where $i \in \{1, \dots n\}$, $r_i + s_i > 0$, $c(r_i, s_i)$ are some numerical coefficients determined from these numbers, and the term in brackets is a nested commutator of $X$'s and $Y$'s. In our case

$$X = e^w(\tilde{L}_{-1} + \eta \tilde{J}_{-1}), \quad Y = -e^w \tilde{L}_{-1}, \tag{A.11}$$

so $[X, Y] = \eta e^{2w} \tilde{J}_{-2}$, and all further commutators with either $X$ or $Y$ will decrease the level of $\tilde{J}_{-n}$ by one and add a multiplicative factor of $e^w$, times some numerical coefficient. Consequently, the term on the right-hand side must take the form

$$e^{e^w(\tilde{L}_{-1} + \eta \tilde{J}_{-1})} e^{-e^w \tilde{L}_{-1}} = e^{\eta \sum_{n=1}^{\infty} c_n e^{nw} \tilde{J}_{-n}}, \tag{A.12}$$

for some numerical coefficients $c_n$ that we will now determine. This can be done by comparing the two ways of computing the $\langle \mathcal{O}(w_1) \mathcal{O}(w_2) \rangle$ overlap presented in section 2.3.

One way consists of simply repeating the steps in (2.42), using (A.12) instead

$$
\begin{aligned}
\langle \mathcal{O}(w_1) \mathcal{O}(w_2) \rangle &= e^{-hw_{12}} \langle h_\lambda | e^{-w_1(\tilde{L}_1 + \eta_{\mathcal{O}} \tilde{J}_1)} e^{w_2(\tilde{L}_{-1} + \eta_{\mathcal{O}} \tilde{J}_{-1})} | h_\lambda \rangle \\
&= e^{w_{12}(\tilde{h} - h)} \langle \tilde{\mathcal{O}}(w_1) e^{\eta_{\mathcal{O}} \sum_{n=1}^{\infty} c_n e^{-nw_1} \tilde{J}_n} e^{\eta_{\mathcal{O}} \sum_{n=1}^{\infty} c_n e^{nw_2} \tilde{J}_{-n}} \tilde{\mathcal{O}}(w_2) \rangle \\
&= e^{w_{12}(\tilde{h} - h)} e^{\frac{k\eta_{\mathcal{O}}^2}{2} \sum_n n c_n^2 e^{-nw_{12}}} \langle \tilde{\mathcal{O}}(w_1) e^{\eta_{\mathcal{O}} \sum_{n=1}^{\infty} c_n e^{nw_2} \tilde{J}_{-n}} e^{\eta_{\mathcal{O}} \sum_{n=1}^{\infty} c_n e^{-nw_1} \tilde{J}_n} \tilde{\mathcal{O}}(w_2) \rangle \\
&= e^{w_{12}(\tilde{h} - h)} e^{\frac{k\eta_{\mathcal{O}}^2}{2} \sum_n n c_n^2 e^{-nw_{12}} + 2\eta_{\mathcal{O}} \tilde{q} \sum_n c_n e^{-nw_{12}}} \langle \tilde{\mathcal{O}}(w_1) \tilde{\mathcal{O}}(w_2) \rangle.
\end{aligned} \tag{A.13}
$$

However, the alternate computation we performed, using just the commutation relations of the unflowed generators, yields directly (2.39). Comparing the two expressions, we conclude that $c_n = \frac{1}{n}$.

# B $\quad J^1 \wedge J^2$ **- deformed free bosons**

In this appendix, we work out in detail the case of $J^1 \wedge J^2$ - deformed free bosons, which should help concretize the general analysis of section 2 . Parts of this analysis have previously appeared in [25, 37].

## B.1 Classical analysis

We start with a variation on the calculation in appendix A of [25]. Consider the action

$$S = -\kappa \int dx^+ dx^- [\partial \phi_1 \bar{\partial} \phi_1 + \partial \phi_2 \bar{\partial} \phi_2 - \lambda(\partial \phi_1 \bar{\partial} \phi_2 - \bar{\partial} \phi_1 \partial \phi_2)] = \int d\sigma dt \, \mathcal{L}, \tag{B.1}$$

where $x^{\pm} = \sigma \pm t$ and $\partial, \bar{\partial} = \frac{1}{2}(\partial_{\sigma} \pm \partial_t)$. The components of the two conserved shift currents $J^{\alpha, a} = -\frac{\partial \mathcal{L}}{\partial(\partial_{\alpha} \phi_a)}$ are

$$J_+^1 = \kappa(\partial \phi_1 + \lambda \partial \phi_2), \quad J_-^1 = \kappa(\bar{\partial} \phi_1 - \lambda \bar{\partial} \phi_2), \tag{B.2}$$

$$J_+^2 = \kappa(\partial \phi_2 - \lambda \partial \phi_1), \quad J_-^2 = \kappa(\bar{\partial} \phi_2 + \lambda \bar{\partial} \phi_1). \tag{B.3}$$

Note that the action satisfies a flow equation of the Smirnov-Zamolodchikov type[22], $\partial_{\lambda} \mathcal{L} = -\epsilon^{\alpha\beta} J_{\alpha}^1 J_{\beta}^2$, provided we choose

$$\kappa = \frac{1}{1 + \lambda^2}. \tag{B.4}$$

Using this, we can construct chiral currents by taking linear combinations of $J_{\alpha}^{1,2}$ and the components of the topologically conserved current $\tilde{J}^a$, with

$$\tilde{J}_+^a = \partial \phi_a, \qquad \tilde{J}_-^a = -\bar{\partial} \phi_a. \tag{B.5}$$

A basis for these currents is

$$J_L^1 = \frac{J^1 + \kappa \tilde{J}^1 - \lambda \kappa \tilde{J}_2}{2} = \kappa \partial \phi_1, \quad J_R^1 = \frac{J^1 - \kappa \tilde{J}^1 - \lambda \kappa \tilde{J}_2}{2} = \kappa \bar{\partial} \phi_1, \tag{B.6}$$

$$J_L^2 = \frac{J^2 + \kappa \tilde{J}^2 + \lambda \kappa \tilde{J}_1}{2} = \kappa \partial \phi_2, \quad J_R^2 = \frac{J^2 - \kappa \tilde{J}^2 + \lambda \kappa \tilde{J}_1}{2} = \kappa \bar{\partial} \phi_2. \tag{B.7}$$

We would now like to compute the Poisson brackets of the chiral currents. For this, we work out the canonical momenta

$$\pi_1 = \kappa(\dot{\phi}_1 + \lambda \partial_{\sigma} \phi_2), \qquad \pi_2 = \kappa(\dot{\phi}_2 - \lambda \partial_{\sigma} \phi_1), \tag{B.8}$$

which satisfy the canonical equal-time commutation relations $\{\phi_a(\sigma), \pi_b(\sigma')\} = \delta_{ab}\delta(\sigma - \sigma')$, and express the currents in terms of them. We find that the Poisson brackets of the above chiral currents are diagonal in this basis, but are proportional to a factor of $\kappa$, which represents the level of the chiral algebra. It is desirable to work instead with the combinations

$$\mathcal{J}_{L,R}^1 = J_{L,R}^1 \pm \lambda J_{L,R}^2, \qquad \mathcal{J}_{L,R}^2 = J_{L,R}^2 \mp \lambda J_{L,R}^1, \tag{B.9}$$

which have level one. Their expression in terms of the canonical variables is

$$\mathcal{J}_{L,R}^1 = \frac{1}{2}(\pi_1 \pm \phi_1' \pm \lambda \pi_2), \qquad \mathcal{J}_{L,R}^2 = \frac{1}{2}(\pi_2 \pm \phi_2' \mp \lambda \pi_1). \tag{B.10}$$

The Hamiltonian density is given by

$$\mathcal{H} = \pi_a \dot{\phi}_a - \mathcal{L} = \frac{1 + \lambda^2}{2}(\pi_1^2 + \pi_2^2) + \frac{1}{2}(\phi_1'^2 + \phi_2'^2) + \lambda(\phi_1' \pi_2 - \phi_2' \pi_1), \tag{B.11}$$

in agreement with our general result (2.6).

---

[22]Our conventions are $\epsilon^{\sigma t} = \epsilon_{t\sigma} = 1$.

## B.2 Quantum analysis

The shift in the chiral charges and the energies of (primary) states on the cylinder with momenta $n^a$ and windings $w_a$ were worked out in full generality in section 2.1. Using the state-operator correspondence, these states correspond to (primary) vertex operators that carry these charges, of the form

$$\mathcal{V}(z,\bar{z}) = \; : e^{ic_L^a \phi_L^a(z) + ic_R^a \phi_R^a(\bar{z})} : \; , \tag{B.12}$$

where $\phi_{L,R}^a$ are the left- and right-moving pieces of the above scalar fields and $c_{L,R}^a$ are coefficients that we would like to determine. For simplicity, we will concentrate on the left-moving piece of the vertex operator and drop the '$L$' index. The action (B.1) implies that the OPEs of the scalars is

$$\phi_a(z)\phi_b(z') = -\frac{\delta_{ab}}{2\kappa}\ln(z-z'). \tag{B.13}$$

The currents $\partial\phi_a$ are primary with dimension 1, as can be checked by computing their OPE with the stress tensor, $T_{zz} = \kappa\sum_a(\partial\phi_a)^2$. The OPE of the above vertex operator with the chiral left currents $\mathcal{J}^a = \kappa(\partial\phi^a + \lambda\epsilon^{ab}\partial\phi_b)$ is thus

$$\mathcal{J}^a(z)\mathcal{V}(0) \sim \frac{c^a + \lambda\epsilon^{ab}c_b}{2z}\mathcal{V}(0). \tag{B.14}$$

Equating the coefficients of $z^{-1}$ with the flowed charge $q^a$ in (2.18), we find

$$c^a = 2\kappa(q^a - \lambda\epsilon^{ab}q_b). \tag{B.15}$$

For the right-moving coefficients, the sign of $\lambda$ is switched and $q^a \to \bar{q}^a$. The associated dimensions are given by the (primary) OPEs

$$T(z)\mathcal{V}(0) \sim \frac{1}{4z^2}\sum_a\frac{c_a^2}{\kappa}\mathcal{V}(0) + \frac{1}{z}\partial\mathcal{V}(0) = \frac{\sum_a q_a^2}{z^2}\mathcal{V}(0) + \frac{1}{z}\partial\mathcal{V}(0). \tag{B.16}$$

The left conformal dimension of this operator is $\sum_a(q^a)^2$, in perfect agreement with (2.19). It is useful to rewrite the exponent as

$$c^a\phi_a = 2q^a\kappa(\phi_a + \lambda\epsilon_{ab}\phi^b) \equiv 2q^a\varphi_a \tag{B.17}$$

and similarly on the right, where $\varphi_{L,R}^a$ simply correspond to the bosonisation of $\mathcal{J}_{L,R}^a$. The vertex operator thus takes the form

$$\mathcal{V}(z,\bar{z}) = \; : e^{2iq^a\varphi_L^a(z) + 2i\bar{q}^a\varphi_R^a(\bar{z})} : \; . \tag{B.18}$$

The mode expansion of $\varphi_L^a$ is given in terms of the modes $J_n^a$ of $\mathcal{J}^a$, i.e.

$$\varphi_L^a(z) = \varphi_L^{a,0} - iJ_0^a\ln z + i\sum_{n\neq 0}\frac{\tilde{J}_n^a}{n}z^{-n} \tag{B.19}$$

and similarly on the right, where we used the fact that $J_n^a = \tilde{J}_n^a$ for $n\neq 0$ to render the expression more familiar. Note that the zero modes of the left and right chiral bosons are independent. It is easy to see that these operators, even inserted at zero, will flow with $\lambda$. To completely specify them, we need to spell out the normal ordering - that is, we put all the annihilation operators to the right of the creation ones

$$\mathcal{V}(z) =: e^{2iq_a\varphi_L^a} := e^{2iq_a\varphi_L^{0,a} + 2q_a\sum_{n=1}^{\infty}\frac{\tilde{J}_{-n}^a}{n}z^n}e^{2q_aJ_0^a\ln z - 2q_a\sum_{n=1}^{\infty}\frac{J_n^a}{n}z^{-n}}. \tag{B.20}$$

The OPE of two such operators can be computed using the BCH formula and the current commutation relations. We find

$$\mathcal{V}_{q_1}(z_1)\mathcal{V}_{q_2}(z_2) = e^{4iq_1^a q_2^b[J_0^a, \varphi_0^b]\ln z_1 - 2q_1^a q_2^a \sum_{n=1}^{\infty} \frac{1}{n}(z_2/z_1)^n} : \mathcal{V}_{q_1}(z_1)\mathcal{V}_{q_2}(z_2) := e^{2q_1^a q_2^a(\ln z_1 + \ln(1-z_2/z_1))} \cdot$$

$$: \mathcal{V}_{q_1}(z_1)\mathcal{V}_{q_2}(z_2) := (z_1 - z_2)^{2q_1 \cdot q_2} : \mathcal{V}_{q_1}(z_1)\mathcal{V}_{q_2}(z_2) :, \tag{B.21}$$

where we used $[\varphi^{0,a}, J_0^b] = \frac{i}{2}\delta^{ab}$. This yields the correct OPE of vertex operators. Note that the zero mode contribution plays an essential role in rendering the correlator translationally-invariant. Including the right-moving piece, we also see that these operators are mutually local, since $q_1^a q_2^a - \bar{q}_1^a \bar{q}_2^a$ is $\lambda$-independent.

We would now like to construct the free boson realisation of the flowed operator $\tilde{\mathcal{V}}(z)$ and compare it to the above operator. At $\lambda = 0$, this operator is simply $\mathcal{V}_{CFT}(z) =: e^{2i\tilde{q}_a \phi_a(z)} :$, where $\phi_a(z)$ has a decomposition of the form (B.19) in terms of the *undeformed* current modes. At finite $\lambda$, $\tilde{\mathcal{V}}$ is given by integrating the flow equation (2.48). The flow operator in this theory is (2.27), which in terms of Fourier modes reads

$$\mathcal{X}_{J\bar{J}} = \sum_{n \neq 0} \frac{1}{2\pi n}(J_n^1 J_{-n}^2 - \bar{J}_n^1 \bar{J}_{-n}^2 + J_n^1 \bar{J}_n^2 - \bar{J}_n^1 J_n^2). \tag{B.22}$$

As one can see from (2.30), the action of this operator on the non-zero modes $\phi_a^{nzm}(z)$ inside $\mathcal{V}_{CFT}$ is to simply turn them into modes of the deformed current, $\tilde{J}_n$. On the other hand, since the zero mode of the current - which equals the $\lambda$ - independent $\tilde{J}_0^a$ - commutes with $\mathcal{X}_{J\bar{J}}$, it will not flow. The scalar zero mode part cannot be inferred from the simple classical flow equation; however, we do know it should be simply $2iq_a \varphi_0^a$, because the flowed state carries charge $q_a$, and the charge is carried entirely by the zero mode. Given all this, a candidate $\tilde{\mathcal{V}}(z)$ operator is

$$\tilde{\mathcal{V}}(z) =: e^{2iq_a \varphi_0^a + 2i\tilde{q}_a(\varphi_{nzm}^a(z) - i\tilde{J}_0^a \ln z)} : . \tag{B.23}$$

This is in perfect agreement with the relation (2.51) between the primary and the flowed operator, upon conformally transforming from $w$ to $z$. The fact that the coefficients of the zero and non-zero modes are different clearly indicates that $\tilde{\mathcal{V}}(z)$ is not a local operator. On the other hand, if we compute the OPE of two such operators, we find

$$\tilde{\mathcal{V}}(z_1)\tilde{\mathcal{V}}(z_2) = \left(1 - \frac{z_2}{z_1}\right)^{2\tilde{q}_1 \cdot \tilde{q}_2} e^{4i\tilde{q}_1^a \ln z_1 q_2^b[\tilde{J}_0^a, q_2^b \varphi_{L,0}^b + \bar{q}_2^b \varphi_{R,0}^b]} : \tilde{\mathcal{V}}(z_1)\tilde{\mathcal{V}}(z_2)$$

$$:= (z_1 - z_2)^{\frac{1}{2}\tilde{q}_1 \cdot \tilde{q}_2} : \tilde{\mathcal{V}}(z_1)\tilde{\mathcal{V}}(z_2) :, \tag{B.24}$$

where we used the fact that the commutator $[\tilde{J}_0^a, \varphi_{L/R,0}^b] = -\frac{i}{2}(\delta^{ab}\delta_L - \frac{\lambda}{2}\epsilon^{ab})$ and that $q^a + \bar{q}^a = n^a$. Therefore, these operators have the same OPE as the original CFT local operators, even though they are not themselves local.

## C  The unflowed $J\bar{T}$ algebra

The algebra of the unflowed generators in $J\bar{T}$ - deformed CFTs was spelled out in [11], and is rather involved. For the purposes of this article, we would like to only collect its subalgebra that contains the global $SL(2, \mathbb{R})$ generators $L_{\pm 1,0}$ and their right-moving counterparts $\bar{L}_{\pm 1,0}$. Unlike the case of standard CFTs, here the algebra generated by these elements does not close: instead, one must include at least the entire infinite tower of left- and right-moving affine $U(1)$ generators. In this appendix, we spell out explicitly the commutation relations of this

subalgebra, which may be helpful in following the main text. Note that, following [11], we use the notation $K_n$ instead of $J_n$ for the current modes.

The commutation relations of the unflowed generators in the deformed $SL(2,\mathbb{R})$ - Kac-Moody subsector are, starting with the right-moving sector

$$[\bar{L}_1, \bar{L}_{-1}] = 2\hbar \bar{L}_0 + \lambda(\bar{K}_{-1}\bar{L}_1 + \bar{L}_{-1}\bar{K}_1 + \lambda \bar{K}_{-1}\bar{K}_1\alpha_1)\alpha_1\,, \quad [\bar{L}_0, \bar{L}_{\pm 1}] = -\bar{L}_{\pm 1}\alpha_{\pm 1}R_v\,, \quad \text{(C.1)}$$

where

$$\alpha_n = \frac{2}{k\lambda^2}\left(\sqrt{(R - \lambda Q_K)^2 + \hbar k\lambda^2 n} - (R - \lambda Q_K)\right) = \frac{n\hbar}{R - \lambda Q_K} + \mathcal{O}(\hbar^2)\,, \quad \text{(C.2)}$$

with $Q_K = J_0 + \frac{\lambda k}{2}H_R$. Note the first commutator implies that the Kac-Moody tower cannot be decoupled, since it is generated by $K_{\pm 1}$ and $L_{\pm 1}$. Then

$$[\bar{K}_m, \bar{K}_n] = \frac{km\hbar}{2}\delta_{m+n} - \frac{\lambda k}{2}\bar{K}_n\alpha_n\delta_{m,0} + \frac{\lambda k}{2}\bar{K}_m\alpha_m\delta_{n,0}\,, \quad \text{(C.3)}$$

$$[\bar{L}_0, \bar{K}_n] = -\bar{K}_n\alpha_n R_v\,, \quad [\bar{L}_{-1}, \bar{K}_n] = -n\hbar\bar{K}_{n-1} - \lambda\bar{K}_{-1}\bar{K}_n\alpha_n + \frac{\lambda k}{2}\bar{L}_{-1}\alpha_{-1}\delta_{n,0}\,, \quad \text{(C.4)}$$

$$[\bar{L}_1, \bar{K}_n] = -n\hbar\bar{K}_{n+1} - \lambda\bar{K}_n\alpha_n\bar{K}_1 + \frac{\lambda k}{2}\bar{L}_1\alpha_1\delta_{n,0}\,. \quad \text{(C.5)}$$

The commutators with the left-moving generators are

$$[L_0, \bar{L}_{\pm 1}] = \bar{L}_{\pm 1}(\pm\hbar - R\alpha_{\pm 1})\,, \quad [L_1, \bar{L}_{\pm 1}] = -\lambda K_1\bar{L}_{\pm 1}\alpha_{\pm 1}\,, \quad [L_{-1}, \bar{L}_{\pm 1}] = -\lambda K_{-1}\bar{L}_{\pm 1}\alpha_{\pm 1}\,, \quad \text{(C.6)}$$

$$[K_m, \bar{L}_n] = -\frac{\lambda k}{2}\bar{L}_n\alpha_n\delta_{m,0}\,, \quad [K_m, \bar{K}_n] = -\frac{\lambda k}{2}\bar{K}_n\alpha_n\delta_{m,0}\,. \quad \text{(C.7)}$$

Note in particular that $L_{-1}$ and $\bar{L}_{-1}$ do not commute at finite $R$. The left generators all commute with $\bar{L}_0$ and their algebra is just the standard $SL(2,\mathbb{R})$ - Kac-Moody one.

The commutation relations of these generators reduce to the standard ones in the $R \to \infty$ limit, given the scaling (C.2) of the $\alpha_n$. In particular, the combinations (3.11) of the right-moving generators do satisfy an $SL(2,\mathbb{R})$ algebra in this limit, though their definition is now only valid up to $\mathcal{O}(1/R)$. The commutators between the right- and the left-moving generators also vanish as $R \to \infty$.

The above commutation relations make it clear that a proposal for the way that $\mathcal{O}(w)$ acts on the cylinder vacuum, of the form suggested in section 2.3

$$\mathcal{O}(w, \bar{w})|0\rangle \overset{?}{=} e^{wh + \bar{w}\bar{h}}e^{e^w L_{-1}}e^{e^{\bar{w}}\bar{L}_{-1}}|h, \bar{h}\rangle \quad \text{(C.8)}$$

misses many of the properties that one may want it to have. First, since $L_{-1}$ and $\bar{L}_{-1}$ do not commute, this is not equivalent to e.g. $\exp\left(e^w L_{-1} + e^{\bar{w}}\bar{L}_{-1}\right)$, and thus the definition of the in/out operator is ambiguous. Second, since

$$e^{\alpha\bar{L}_0}e^{e^{\bar{w}}\bar{L}_{-1}}e^{-\alpha\bar{L}_0} = e^{e^{\bar{w} + \alpha R_v\alpha_1^r}\bar{L}_{-1}}\,, \quad \text{(C.9)}$$

we see that $\bar{L}_0$, despite being the unambiguous generator of right-moving translations on the cylinder, does not induce a simple shift in the label $\bar{w}$ of the operator proposed above; rather, the shift is field-dependent, except in the $R \to \infty$ limit.

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
