# Peer review of "A definition of primary operators in $J\bar T$-deformed CFTs"

_SciPost Physics, doi:SciPost Phys. 13, 045 (2022)_

## Round 2 · Referee Report · Anonymous (Referee 1) · 2022-3-9

Strengths

1) New definition of operators in the $J{\bar T}$-deformed CFTs, which satisfy nice properties under the symmetry algebra 2) Computation of correlators of these operators on the plane 3) Clearly written with a detailed review of the well-understood $J{\bar J}$-deformation that motivates the construction

Weaknesses

1) The definition of the operators does not follow from any first principles, though it satisfies many nice properties 2) The computations are done in a mixed formalism which starts from the cylinder but does the computations only in the infinite radius limit, it would be nicer to either compute precisely on the cylinder or to phrase everything directly on the plane

Report

The paper suggests a definition of operators in $J{\bar T}$-deformed theories that transform nicely under the symmetry algebra of these theories (that was analyzed in detail in previous works by the author), and computes their correlation functions on the plane, with relatively simple results expressed as a shift in the conformal dimensions of the operators. The paper starts from an analysis of $J{\bar J}$-deformed theories, which are standard CFTs where the operators and correlators are well-understood, but which can also be analyzed in a language that can be generalized for $J{\bar T}$ deformations. The paper is clearly written and the results are nice and reasonable, in particular the advantages and disadvantages of the construction are described very clearly. I am happy to recommend the publication of this interesting paper in SciPost Physics.

Requested changes

None

---

## Round 2 · Referee Report · Anonymous (Referee 2) · 2022-5-10

Strengths

1) The author proposes a definition for the notion of primary operators in the JJbar and JTbar deformed CFTs 2) The author computes some correlation functions of these operators and draws comparisons with the existing literature 3) The article is well structured, clear in its scope, and exhausting in its arguments and derivations

Weaknesses

1) The reader can be easily confused by both the length of certain formulae and the notation which can be somewhat obscure in places

Report

I believe this work satisfies the acceptance criteria of this journal and support its publication.

---

## Editorial Decision

published